

# Quantifying the relationship among PM$_{2.5}$ concentration, visibility and planetary boundary layer height for long–lasting haze and fog–haze mixed events in Beijing city

Tian Luan[1,2], Xueliang Guo[1,2,3], Lijun Guo[1,2], Tianhang Zhang[4]

[1]State Key Laboratory of Severe Weather (LASW), Chinese Academy of Meteorological Sciences, Beijing, 100081, China
[2]Key Laboratory for Cloud Physics, Chinese Academy of Meteorological Sciences, Beijing, 100081, China
[3]Collaborative Innovation Center for Meteorological Disasters Forecast, Early Warning and Assessment, Nanjing University of Information Science and Technology, Nanjing, 210044, China
[4]National Meteorological Center, Beijing, 100081, China

*Correspondence to*: Xueliang Guo (guoxl@camscma.cn)

**Abstract.** The air quality and visibility are strongly influenced by aerosol loading and meteorological conditions. The quantification of their relationships is critical to understanding the physical and chemical processes and forecasting of the polluted events. We investigated and quantified the relationship among PM$_{2.5}$ (particulate matter with aerodynamic diameter is 2.5 μm and less) mass concentration, visibility and planetary boundary layer (PBL) height in this study based on the data obtained from four long–lasting haze events and seven fog–haze mixed events from January 2014 to March 2015 in Beijing city. The data were sampled by the state–of–the–art instruments such as Micro Pulse Lidar (model MPL–4B), particulate monitor (model TEOM 1405–DF), ceilometer (model CL31), visibility sensor (model PWD20) and profiling microwave radiometer (PMWR, model 3000A) as well as some conventional meteorological instruments during the field campaign for haze and fog–haze mixed events in northern China. The statistical results show that there was a negative exponential function between the visibility and the PM$_{2.5}$ mass concentration for both haze and fog–haze mixed events (with the same R$^2$ of 0.80). However, the fog–haze events caused a more obvious decrease of visibility than that for haze events due to the formation of fog droplets that could induce higher light extinction. The PM$_{2.5}$ concentration had inversely linear correlation with PBL height for haze events and negative exponential correlation for fog–haze mixed events, indicating that the PM$_{2.5}$ concentration is more sensitive to PBL height in fog–haze mixed events. The visibility had positively linear correlation with the PBL height with the R$^2$ of 0.35 in haze events and positive exponential correlation with the R$^2$ of 0.55 in fog–haze mixed events. We also investigated the physical mechanism responsible for these relationships among visibility, PM$_{2.5}$ concentration and PBL height through typical haze and fog–haze mixed event, and found that a double inversion layer formed in both typical events and played critical roles in maintaining and enhancing the long–lasting polluted events. The upper–level stable inversion layer formed by the persistent southwest warm and humid airflow caused the PM$_{2.5}$ accumulation and subsequent surface cooling as well as the formation of a weak low–level inversion layer. The formation of low–level inversion layer further enhanced the PM$_{2.5}$ accumulation and surface cooling process, and induced a strong descending process of the upper–level inversion layer with warm and humid air, which significantly strengthened the PBL stability and formed a deep stable PBL in the daytime,





and in return rapidly increased the PM$_{2.5}$ concentration. This positive feedback was particularly strong when the PM$_{2.5}$ mass concentration was larger than 150–200 µg m$^{-3}$. Therefore, the formation and subsequent descending processes of the upper–level inversion layer should be an important factor in maintaining and strengthening the long–lasting severe polluted events, which has not been revealed in previous publications. The feedback caused an obvious and more rapid increase of PM$_{2.5}$

concentration and a significant deterioration of air quality and visibility in fog–haze mixed events.

## 1 Introduction

Due to a rapid economic development, the haze and fog events characterized by the high fine particulate matter (i.e. PM$_{2.5}$, particulate matter with aerodynamic diameter is 2.5 µm and less) levels have occurred during the last few decades in China, especially in the most developed and high–populated city (Chan and Yao, 2008; Zhang et al., 2013; Huang et al., 2014; Zhang

and Cao, 2015). For instance, in January 2013, Beijing (and the entire inland China) suffered from the extremely severe and persistent haze and fog pollution, registering the highest PM$_{2.5}$ hourly concentration of 886 µg m$^{-3}$ (Zhang et al., 2013; Wang et al., 2014a; Zhang et al., 2014). The high frequency of extremely severe and persistent haze and fog events in China leads to a high public concern due to its poor visibility and adverse health effects (Tie et al., 2009; Chen et al., 2013; Pope and Dockery, 2013).

Aerosol particles and fog droplets are responsible for the reduction of visibility by scattering and absorbing light, according to their number and properties, such as size, shape, and chemical composition. Atmospheric humidity is a major factor affecting the particle properties, as aerosols can grow by the uptake of water. When relative humidity (RH) is larger than 95 %, the atmospheric visibility can be critically reduced (Chen et al., 2012). When RH is larger than 100 %, some of the hygroscopic aerosols can be activated and form fog droplets (Pruppacher and Klett, 1978). The sudden increase in particle size causes a

sharp drop in visibility, usually to distances below 1 km (Elias et al., 2009). Baumer et al. (2008) found that the visibility decrease was associated with a continuous increase in the number size distribution of particles with diameters larger than 300 nm in South–West Germany. Particles have grown into a size interval in which their diameter is of the same order as the wavelength of the visible light, which leads to a more effective light scattering. Therefore, the visibility should decrease.

Large emission sources emit primary aerosols and the precursors of secondary aerosols, resulting in high loads of aerosols (i.e.

sulfate, nitrate, ammonium, black carbon, organic carbon, and dust) (Zhang et al., 2009; Zhang et al., 2012; Zhang et al., 2013). This is the main reason for the deterioration of visibility and frequent haze events through light extinction (Cao et al., 2012; Han et al., 2016). During the haze periods, the concentration of particulate matter is much higher than that on normal days, and fine–mode aerosols are predominant (Quan et al., 2011; Zhang and Cao, 2015). Han et al. (2016) found that the 71±17 % of PM$_{10}$ was PM$_{2.5}$ in Beijing and the increasing of PM$_{2.5}$ contributed to visibility impairment significantly.

Fog and haze events usually occur in the stable PBL, which is located at the lowest atmospheric layer and strongly influenced by the exchange of momentum, heat, and water vapor at the earth's surface. Many previous publications showed that fog and haze events were usually formed in a weak high–pressure system with low surface wind, which was unfavourable for air



mixing and pollutants diffusion (Liu et al., 2007; Kang et al., 2013; Zhao et al., 2013b; Zheng et al., 2015b). The aerosols directly emitted from the surface and secondly formed aerosols were concentrated in the PBL, resulting in high concentrations inside of the surface. Sun et al. (2013) suggested that the $PM_{2.5}$ distribution depicted a 'higher–bottom and lower–top' pattern based on the observation of the 325 m tower in Beijing. Zhang and Cao (2015) showed that the $PM_{2.5}$ concentration at night

was about 2 times higher than that in the afternoon. The lowest concentrations were observed in afternoon hours when the PBL height became larger and wind speed increased. Many studies also found that the diurnal variation of the pollutants was anti–correlated with the diurnal variation of PBL height (Chou et al., 2007; Boyouk et al., 2010). Yang et al. (2013) indicated that one of the possible factors leading to the deteriorated air quality in Hong Kong was the decreasing trend of the daily maximum of mixing layer height based on 6.5 y measurements.

Interactions among aerosols, radiation and atmospheric boundary layer structure are very complex processes and still uncertain in many aspects. Aerosols such as black carbon can strongly absorb solar radiation and modify the vertical profile of temperature in the atmosphere and stabilize the PBL structure (Ding et al., 2016). The accumulation of aerosols in the PBL can lead to a more stable atmosphere. Analysis of a heavy pollution episode in fall 2004 over northern China showed that the instantaneous irradiance at the surface decreased by about 350 W m$^{-2}$ and the atmospheric solar heating was about 300 Wm$^-$

$^2$; therefore, a more stable atmosphere was expected (Liu et al., 2007). Quan et al. (2013) found that the heat flux of surface and PBL height in haze condition were significantly lower than that under clear sky condition, and proposed that the feedback might exist between PBL height and aerosol loading. The enhancement of aerosols tends to depress the development of PBL by decreasing solar radiation, while the repressed structure of PBL will in turn weak the diffusion of pollutants, leading to the heavy pollution.

The model results from Gao et al. (2015) showed that during the fog and haze mixed event over the North China Plain, aerosols led to a significant negative radiation forcing at the surface and a large positive radiation forcing in the atmosphere and induced significant changes in meteorological variables in day time. As a result, atmosphere was much more stable and thus the surface wind speed decreased and the PBL height decreased. The maximum increase of hourly surface $PM_{2.5}$ concentration is 50 μg m$^{-3}$ over Beijing. Wang et al. (2014b) implied that the interaction between aerosol and radiation played an important role in

the haze episode in January 2013 from simulated results by WRF–CMAQ. Petäjä et al. (2016) showed that aerosol–boundary layer feedback remained moderate at fine particular matter concentrations lower than 200 μg m$^{-3}$, but that it became increasingly effective at higher particular matter loadings resulting from the combined effect of high surface particular emissions and massive secondary particular matter production within boundary layer.

The influence of convective mixing on the air quality has been recognized for decades, yet data showing this phenomenon

remain rather limited for the lack of temporal resolution in the PBL height measurement. The development and application of lidar have made such an investigation possible. The lidar technique has provided a useful tool to investigate cloud and aerosol properties with high temporal resolution in the atmosphere (Welton et al., 2002; Zhang et al., 2015). It is also known to be suitable for studying the PBL and its evolution (Yan et al., 2013; Zhang et al., 2016). Micro Pulse Lidar (MPL) systems were used to measure aerosol properties during the Indian Ocean Experiment (INDOEX) 1999 field phase (Welton et al., 2000;





2002). Chen et al. (2001) observed seasonal changes of the mixing layer height in Japan by using the MPL. He et al. (2008) analysed aerosol vertical distribution in the lower troposphere by examining the aerosol extinction profiles derived from MPL measurements in Hong Kong. Yang et al. (2013) analysed the diurnal, seasonal and inter–annual variation of PBL height from 6.5 y lidar data set over Hong Kong. Zhang et al. (2015) showed the evolution of aerosol vertical distribution and the PBL

height during the haze events in Shanghai.

Although there have been many theoretical and observational studies on the characteristics of $PM_{2.5}$ concentration, visibility and PBL height as well as their correlations, the quantitative investigations on their relationship for long–lasting haze and fog–haze mixed events are few. The feedback mechanism between $PM_{2.5}$ concentration and PBL height has not been well understood. Since 2013, a comprehensive field campaign on haze and fog–haze mixed events had been conducted in northern

China. The state–of–the–art instruments such as Micro Pulse Lidar (MPL), ceilometer (CL31), profiling microwave radiometer (PMWR), and particulate monitor as well as some conventional meteorological instruments have been used in this field campaign. The data were acquired during the long–lasting pollutant events from January 2014 to March 2015. In this study, the relationship among $PM_{2.5}$, visibility and PBL height for both haze and fog–haze mixed events were investigated and quantified. The physical mechanism responsible for their relationships and feedbacks were also investigated and discussed.

## 2 Methodologies

### 2.1 Observational site and instruments

The observational site is located at the campus of China Meteorological Administration (CMA) (39°57′ N, 116°20′ E) in Beijing city, northern China. It was built on the roof of a 20 m tall building from the year of 2013, which located in the northwest part of urban Beijing, close to the West 3th Ring Road, without any major sources nearby. Data used in this study

include the mass concentration of $PM_{2.5}$, visibility and the PBL height obtained from ceilometer, as well as the temperature, RH, water vapor and liquid profiles retrieved by PMWR. The vertical profiles of aerosol in the troposphere, and the PBL height were also obtained from a ground–based MPL installed in a working container 10 m far away from the building. The meteorological parameters (surface wind, ambient temperature and RH) were obtained from Haidian automatic weather station. Radiosonde sounding data (twice a day at 08:00 and 20:00 LST (Local Standard Time)) and daily radiation data were obtained

from Beijing Meteorological station (39°56′N, 116°17′E, 54511). These measurement sites are located within distances of 30 km each other, as shown the sketch map of Fig.1. NCEPT FNL (Final) Operational Global Analysis data were used to analyse the meteorological factors and weather patterns. The data on 1 degree by 1 degree grids prepared operationally every six hours. Micro Pulse Lidar (MPL–4B) (Sigma Space Corporation, USA) is a portable eye–safe elastic backscatter lidar, fully automated. It can work 24 h per day outdoors in an unattended mode under almost any weather conditions. The laser light source is a

diode–pumped frequency–doubled solid–state laser (Nd: $YVO_4$ at 532 nm) yielding pulsed visible green light. The pulse repetition frequency is 2500 Hz. The peak value of the optical energy of laser beam is 10 μJ. The pulse duration was set to 100 ns, and the pulse interval was set to 200 ns, corresponding to a vertical resolution of 30 m. Before processing the lidar data,





the raw signal needed correction including optical overlap, after pulse, dead time, and background noise correction (Campbell et al., 2002; 2008). Then the aerosol optical properties can be retrieved by the lidar equations using Fernald algorithm (Fernald, 1984). The lidar signal is not available in the first hundred meters because of after pulse effect. The lowest sounding height was set to 100 m.

Ceilometer used in this study is a Vaisala CL31 model, described in detail in Münkel and Räsänen (2004) and Münkel et al. (2007). In brief, CL31 is equipped with an indium gallium arsenide/ metal–organic chemical vapor deposition (InGaAs/MOCVD) pulsed diode laser emitting at $905\pm10$ nm and having an energy per pulse of 1.2 µJ$\pm$20 %. The emission frequency is 10 kHz, while the pulse duration is 110 ns. According to the Vaisala ceilometer CL31 User's Guide (2009), the full overlap of the instrument is achieved for altitudes higher than 10 m, although in practice on the order of 70 m (Martucci
et al., 2010). The attenuated backscatter coefficient is obtained from 10 m to 7.5 km height, with a selectable spatial resolution of 5 or 10 m and temporal resolution of 2 s to 120 s. In this study, we used 10 m raw range resolution and 16 s raw temporal resolution. This temporal resolution was deemed sufficient for analysing the development of PBL structure. Apart from the very strong backscatter from clouds and fogs, the weaker gradients of the backscatter intensity were mainly determined by the number and the size spectrum of aerosol particles suspending in the air.

TEOM 1405–DF particulate monitor (Tapered Element Oscillating Microbalance, Thermo Fisher Scientific, USA) was used to continuously measure particulate matter mass concentration. According to the TEOM 1405–DF Operating Guide (2009), the monitor draws ambient air through two TEOM filters at constant flow rate, continuously weighing the filters and calculating near real–time mass concentrations of both $PM_{2.5}$ and PMcoarse (2.5 µm <particulate matter with aerodynamic diameter <=10 µm). By adding these two values, the $PM_{10}$ mass concentration is obtained. The FDMS unit dries the sample flow and
automatically generates mass concentration measurement that account for both nonvolatile and volatile particulate matter components. The volatile faction of the collected sample is automatically compensated by using a switching valve to change the path of the fine and coarse sample flows every 6 min. The filters were replaced when the filter loading percentage nears 100 %.

Profiling microwave radiometer (PMWR–3069A, Radiometrics Co., USA) collects atmospheric radiation measurements in
the 20 to 200 GHz region to retrieve temperature, RH, water vapor and liquid profiles. The temperature profiling subsystem utilizes sky brightness temperature observations at selected frequencies between 51 and 59 GHz. The water vapor profiling subsystem utilizes sky brightness temperature observations at selected frequencies between 22 and 30 GHz. The water vapor channels are calibrated by means of tipping curves; the temperature channels are calibrated by a liquid nitrogen cold target. The temperature, RH, vapor density and liquid water content (LWC) profiles in this study are retrieved from PMWR
measurements at zenith direction. The temporal resolution of 2–3 min, and the vertical resolutions of 50 m from the surface to 500 m, 100 m from 500 m to 2 km, and 250 m from 2 km to 10 km. The accuracy of MWR profiles is compatible with most meteorology applications, especially in the lower troposphere (Cimini et al., 2011; Ware et al., 2013; Gultepe et al., 2015).

The visibility sensor (PWD20, Vaisala Co., Finland) with range of 10–20,000 m was employed to monitor atmospheric visibility.



## 2.2 Determination of the PBL height

The PBL height is determined at the altitude where a sudden decrease in the scattering coefficient occurs. In this paper, we used the wavelet covariance transform method by Brooks (2003) to inverse the PBL height by MPL, which is based on scanning the backscatter profile with a localized impulse function and maximizing the covariance between the backscatter profile and impulse function. A step function Haar wavelet is defined as Eq. (1):

$$h\left(\frac{z-b}{a}\right) = \begin{cases} +1: & b - \frac{a}{2} \leq z \leq b \\ -1: & b \leq z \leq b + \frac{a}{2} \\ 0: & elsewhere \end{cases} \tag{1}$$

where $z$ is altitude, $b$ is the location at which the Harr function is centered, and $a$ is the spatial extent of the function. The covariance transformation of the Haar function ($W_f$) is calculated by Eq. (2):

$$W_f(a,b) = \frac{1}{a} \int_{z_b}^{z_t} f(z) h\left(\frac{z-b}{a}\right) dz \tag{2}$$

where $f(z)$ is the signal of the MPL backscatter profile, and $z_b$ and $z_t$ are the lower and upper limits of the profile. A local maximum in $W_f(a,b)$ identifies a step in $f(z)$, located at $z=b$. $b$ is the top of the PBL.

Ceilometer is a robust low–power, low–cost and low–maintenance Lidar designed to determine the cloud base height but also provide the backscatter profile, though with less sensitivity than a Lidar. Several studies have proposed that ceilometer–measured backscatter profiles can be used to derive the PBL height (Eresmaa et al., 2006; Münkel, 2007; Eresmaa et al., 2012; Haeffelin et al., 2012; Schween et al., 2014). "Structure of the atmosphere" (STRAT–2D) algorithm was selected for estimating the PBL height, which was proposed in the literatures (Morille et al., 2007; Haeffelin et al., 2012; Wiegner et al., 2014). To eliminate the influence of inherent noise and aerosol layers on the data, spatial and temporal averaging must be carried out before the gradient method used to calculate the PBL height. In this paper, the settings use 80 m height averaging close to the ground. This interval is gradually increased to 360 m averaging used at heights above 1500 m. Time averaging is dependent on the current signal noise. It varies between 14 min during nighttime and 52 min on a bright, cloudy day (Tang et al., 2016). This method uses the vertical aerosol backscatter gradient, whereby strong negative gradients can indicate the PBL height. STRAT–2D determines three candidates for PBL height: the largest, the second largest gradient and the lowest height gradient.

## 2.3 Data processing

In this work, the criterions of long–lasting haze and fog–haze mixed events were defined as: (1) for the long–lasting haze events, the following conditions should be satisfied: 1 km < hourly visibility and minimum visibility <=10 km and the hourly PM$_{2.5}$ concentration >=50 µg m$^{-3}$ for lasting more than 72 hours continuously. (2) for the long–lasting fog–haze mixed events, the following conditions need to be satisfied: minimum visibility <=1 km, hourly visibility <=10 km and the hourly PM$_{2.5}$ concentration >=50 µg m$^{-3}$ for lasting more than 72 hours continuously. From January 2014 to March 2015, the total number of persistent pollutant cases is 11, in which four haze cases and seven fog–haze mixed cases were obtained. Table 1 listed all cases investigated.



## 3 Results and discussion

### 3.1 Relationship between visibility and PM$_{2.5}$ mass concentration

Previous studies indicated that the increase in PM$_{2.5}$ mass concentration contributed to visibility impairment significantly in China (Cao et al., 2012; Han et al., 2013; Zhao et al., 2013a; Li et al., 2015; Han et al., 2016). The relationships between

5 visibility and PM$_{2.5}$ mass concentrations for both long–lasting haze and fog–haze mixed events are shown in Fig.2, and the corresponding regression results are given in Table 2. It shows that there was a negative exponential function between the visibility and the PM$_{2.5}$ mass concentration for both haze and fog–haze mixed events with the same R$^2$ of 0.80. The relationship for haze events is consistent with the previous result of Han's study on the relationship between daily averaged PM$_{2.5}$ concentration and visibility under stable meteorological condition from October 2013 to September 2014 at Beijing. However,

the fog–haze mixed events could cause greater visibility impairments, for example, in the haze events, the visibility reduced from 5.8 to 2.7 km as the PM$_{2.5}$ concentration increased from 100 to 200 μg m$^{-3}$, and in the fog–haze mixed events the visibility reduced from 4.7 to 1.5 km for the same amount of PM$_{2.5}$ concentration increase. The differences between the two conditions are mainly due to the increase of RH and the formation of fog droplets that could induce higher light extinction. The averaged RH observed by Haidian automatic weather station in haze and haze–fog mixed events is 46.7 % and 74.6 %, respectively.

Under high RH conditions, a large amount of water vapor coated on water–soluble particle surface, enlarges the particle size, which significantly enhances the particulate light scattering efficiency and deteriorates visibility. When water vapor saturated, haze aerosols could be activated to form fog droplets, which led to the further decrease of visibility (Elias et al., 2009; Klein and Dabas, 2014; Guo et al., 2015).

Moreover, in the aqueous phase, the production rate of sulfate and nitrate aerosols was enhanced by aqueous–phase chemistry

(i.e., in–fog oxidation by dissolved ozone (O$_3$) and hydrogen peroxide (H$_2$O$_2$)) (Andreae and Rosenfeld, 2008; Seinfeld and Pandis, 2012) and heterogeneous chemistry (Pandis et al., 1992; Zheng et al., 2015a; Zheng et al., 2015b). Sulfur oxidation ratios showed a rapid increase as a function of RH, which varied from ~0.05 at RH < ~40 % to 0.2 at RH = ~80 % and greatly increased to 0.4 via aqueous–phase processing (Sun et al., 2014). Han et al. (2013) showed that sulfate and nitrate were the two major inorganic aerosol components of PM$_{2.5}$ in Beijing that evidently decreased visibility by contributing 40–45 % to the

total extinction coefficient value. Cao et al. (2012) indicated that high secondary inorganic aerosol contributions (i.e. SO$_4^{2-}$ and NO$_3^-$) were the main contributors for visibility <5 km. Kang et al. (2013) indicated that aerosol concentration in the diameter from 0.6 to 1.4 μm increased dramatically and mainly attributed to the remarkable increase of scattering coefficient and decrease of visibility in a long–lasting haze in Nanjing. Shi et al. (2014) addressed the relationship of visibility with PM$_1$ and total water–soluble ions during the periods of December 2012. They found that hourly total water–soluble ions mass

concentration had a better correlation with visibility, and the formation / dissociation of NH$_4$NO$_3$ and NH$_4$Cl exerted great impacts on visibility. Strong NH$_3$ and HNO$_3$ reaction resulted in the enhancement of NH$_4$NO$_3$ mass fraction under high RH condition that contributed to visibility degradation.



## 3.2 Relationship between PM$_{2.5}$ mass concentration and PBL height

The PBL height can be derived from both MPL and CL31 instruments. Figure 3 shows the PBL height determined by MPL versus that by CL31, and the 1:1 line is given for reference. The figure reveals $R^2=0.70$ with differences between –408 m and 692 m. The number of the difference of PBL height retrieved by MPL and CL31 within ± 300 m accounts for 97 % of the total.

Tsaknakis et al. (2011) obtained that the differences of PBL height derived by Raymetrics lidar and CL31 were about 50–100 m based on the two cases in the midday. This difference may be attributed mainly to the different wavelengths used by MPL and CL31. The PBL height derived by MPL is usually used as a reference in detecting the aerosol vertical distribution by more advanced and powerful lidars. It shows in Fig.3 that CL31 underestimated at low PBL height and overestimated at high PBL height. Thus, the PBL height derived from the MPL was used in the following part of this paper. The PBL heights in pollution

condition varied from 150 m to 1000 m and the heights under 500 m accounts for 87 %.

It should be noted that since MPL determines the PBL height by measuring the attenuated backscatter profile, it cannot calculate PBL height correctly through sudden changes in the attenuated backscatter profiles and results in serious underestimations (Tang et al., 2016). Such as in situation that the strong northerly winds with dry and clear air masses prevail in observation site, the atmospheric aerosols spread rapidly to the downstream region, resulting in a dramatic decrease in local

aerosol concentration and good visibility. Once the aerosol concentration becomes uniform in the vertical direction, the lidar cannot calculate the PBL height correctly through sudden changes in the attenuated backscatter profiles, resulting in a serious underestimation.

The statistical relationship between PM$_{2.5}$ mass concentration and PBL height was investigated and shown in Fig.4. It shows that the PM$_{2.5}$ concentration had inversely linear correlation with the PBL height with the $R^2$ of 0.34 for haze events and

20 negative exponential correlation with the $R^2$ of 0.49 for fog–haze mixed events, indicating that the PM$_{2.5}$ concentration is more sensitive to the PBL height in fog–haze mixed events. The PM$_{2.5}$ concentrations of 50 µg m$^{-3}$, 100 µg m$^{-3}$, 200 µg m$^{-3}$, 300 µg m$^{-3}$, and 400 µg m$^{-3}$, corresponded to the PBL heights of 0.83 km, 0.51 km, 0.33 km, 0.30 km, and 0.29 km, respectively in fog–haze mixed events.

The feedback between PBL height and PM$_{2.5}$ mass concentration was obviously different in the haze events and fog–haze

mixed events. In the haze events the PM$_{2.5}$ mass concentration increased almost linearly with the decrease of PBL height, while in the fog–haze mixed events the PM$_{2.5}$ mass concentration initially tended to show a relatively slow increase with the decrease of PBL height. As long as the PBL decreased to the height below 0.4–0.5 km, the slight decrease of PBL height could cause a rapid increase of PM$_{2.5}$ mass concentration. Petäjä et al. (2016) investigated the haze cases in Nanjing city and pointed out that the aerosol–boundary layer feedback remained moderate at fine particulate matter concentrations lower than about 200 µg m$^{-}$

$^3$, but that it became increasingly effective at higher particulate matter loadings. Our investigation shows that this phenomenon became more obvious in fog–haze mixed event.



### 3.3 Relationship between visibility and PBL height

Theoretically, the relationship between PBL height and atmospheric visibility is not obvious under clean and clear condition. However, under polluted conditions the variation of PBL height may directly cause the variation of aerosol concentration within the PBL, and induce the change of visibility. The statistical results in Fig.5 show that there were strong relationships between visibility and PBL height for both haze and fog–haze mixed events. A positive linear correlation with the $R^2$ of 0.35 existed in haze events and positive exponential correlation with the $R^2$ of 0.55 existed in fog–haze mixed events between visibility and PBL height.

### 3.4 Physical mechanism responsible for the relationship among PM$_{2.5}$, visibility and PBL height

To clarify the physical mechanism responsible for the relationship among PM$_{2.5}$, visibility and PBL height obtained above, two typical cases of long-lasting haze and fog–haze mixed events are presented and further investigated.

#### 3.4.1 Typical haze event

A typical haze event in April 2014 lasting for 74 hours starting at 22:00 LST (Local Standard Time) on 11 April and ending at 23:00 on 14 April during which visibility was less than 10 km. The synoptic situation during the haze event characterized as a col pressure field covering North China Plain. This weather system would lead to calm surface wind and stably stratified atmospheric condition, which was favourable for the accumulation of air pollutants. A cold front passed through Beijing on April 14 and ended the long–lasting haze event.

Figure 6 shows the temporal variations of surface meteorological and environmental factors in the whole process of the haze event. Both air temperature and RH presented a clear diurnal cycle, but they showed a gradually increasing tendency during the haze period due to the persistent southwest warm and humid airflow. The temperature increased from 7.9 ℃ to 25 ℃, with an average of 16.6±5.1 ℃, while the RH was in the range of 28 % to 89 %, with an average of 55±17 %. The variation of RH inversely corresponded that of temperature. The temperature and RH derived from microwave radiometer showed a consistent tendency with those observed by surface automatic weather station. The wind speed varied from 0 m s$^{-1}$ to 3.9 m s$^{-1}$, with an average of 0.8 m s$^{-1}$, suggesting that the horizontal transport of aerosols was very weak. The PM$_{2.5}$ mass concentration was inversely correlated with visibility. PM$_{2.5}$ reached the highest at 12:00 on 14 April with the value of 304 µg m$^{-3}$, corresponding to the hourly mean visibility of 1317 m. PM$_{2.5}$ decreased dramatically after 21:00 on 14 April. The visibility continued to rise until the end of the event. The average PM$_{2.5}$/PM$_{10}$ is as high as 0.82, implying that fine particles were dominant in the atmosphere.

The temporal variation of vertical distributions of temperature, RH, LWC and vapor density retrieved by MWRP during the whole haze event is shown in Fig.7. Many studies demonstrated that PMWR is a useful tool to sense the thermodynamic structure of the lower troposphere continuously by providing profiles of temperature and humidity with reasonable accuracy and height resolution (Ware et al., 2003; Ware et al., 2013; Xu et al., 2015). The inter–comparison with the radiosonde data



demonstrated the good correlation of temperature and vapor density retrievals (Guo and Guo, 2015; Xu et al., 2015). The biases of temperature retrieved by PMWR against radiosondes increased with height, and the maximum of bias is 4 ℃ under 2 km; the bias of water vapor profile was smaller than 1 g m$^{-3}$ (Guo and Guo, 2015). Compared to the radiosonde temperature profiles in Fig.10 (a) and (c), due to the lower vertical resolution, the PMWR could not capture the temperature inversions at the upper level. But the relatively high RH value could be well captured at the height between 0.7 km to 1.6 km, indicating the dominant south-westerly warm and humid airflow during the haze event. The LWC was less than 0.01g m$^{-3}$.

Figure 8 shows the time–height distribution of the backscatter density detected by the CL31 and the normalized relative backscatter (NRB) of MPL, and time evolutions of the MPL–derived PBL height and PM$_{2.5}$ mass concentration during the whole haze event. Compared to the MPL, CL31 could not detect all backscatter of the haze aerosols in fine particles due to the longer lidar wavelength (910 nm). Figure 8 (b) shows that the height indicated by the high value of NRB tended to decrease during the whole haze event, indicating that the PBL height tended to decrease with time evolution until in the end of the haze event. Generally, a negative correlation or negative feedback can be found between PM$_{2.5}$ concentration and PBL height (Fig.8 (c)). However, the feedback between PM$_{2.5}$ concentration and PBL height was relatively weak when the PM$_{2.5}$ concentration was below 200 μg m$^{-3}$. When PM$_{2.5}$ concentration was above 200 μg m$^{-3}$, the negative feedback became strong. For example, before 14 April, the daily averaged PBL height was above 0.4 km and the PM$_{2.5}$ concentration was generally below 200 μg m$^{-3}$. After 14 April, the PBL height rapidly reduced to 0.3 km and then the PM$_{2.5}$ concentration increased its maximum value of 300 μg m$^{-3}$. The interesting phenomenon has similar result recently obtained in Nanjing city in south China (Petäjä et al., 2016). The lowing PBL height compressed the aerosol particles into a shallow vertical layer, and prevented the vertical dispersion of the aerosol particles, leading to an increase in the surface aerosol concentrations, which is consistent to the previous study in the region (Quan et al., 2014).

To reveal the lowing process of the PBL height, the time–pressure distribution of vertical velocities in Beijing from 11 to 14 April 2014 is presented in Fig.9. It indicates that before 13 April, the atmospheric layer from the surface to the middle of troposphere was under a weak updraft condition that was favoured the upward diffusion of aerosols. After 13 April, the downdraft zone started to develop at the upper levels of boundary layer. The formation of this downdraft zone strongly suppressed the upward diffusion of polluted aerosol particles. The PBL height would be lowed, which forced the PM$_{2.5}$ to concentrate at the lower layers and further deteriorated air quality. Therefore, the occurrence of the downdraft zone was one of the important factors to decrease PBL height during the haze event.

The aerosol–boundary layer feedback by blocking solar radiation process was suggested as a plausible explanation for the most severe haze episodes in the regions of China (Ding et al., 2016; Petäjä et al., 2016). However, the mechanism that causes the formation of downdraft zone has not been fully understood due to lack of the direct observational data. Since the formation of the downdraft zone can be caused by many mechanisms such as a cooling process induced by upper–level cold air intrusion or cloud process, or an enhanced long–wave radiation emission at the top of boundary layer due to the high accumulation of aerosols at this level. In the daytime, the growth of the PBL height is strongly depended on the surface solar radiation. In the clear day, the PBL can be fully developed through the solar radiation heating. However, if solar radiation is absorbed or





scattered by aerosol particles or clouds, the PBL cannot be fully developed, and the daytime PBL heights can be significantly reduced (Yu et al., 2002).

Table 3 presents the parameters of radiation from Beijing Meteorological station during the haze event. It shows that the daily total horizontal plane direct radiation was significantly reduced from 7.67 MJ m$^{-2}$ d$^{-1}$ on 13 April to 4.91 MJ m$^{-2}$ d$^{-1}$ on 14 April, and at the same time the amount of total scattering radiation was increased from 11.07 MJ m$^{-2}$ d$^{-1}$ to 12.65 MJ m$^{-2}$ d$^{-1}$. These results demonstrate that aerosol particles played important roles for reducing the solar radiation and inhibited the development of PBL heights during the daytime haze event.

To investigate the influence of surface radiation change on temperature profile and characteristic of temperature and humidity variations in the whole haze event, the profiles of temperature and RH are displayed in Fig. 10. The apparent features of vertical temperature and RH profiles were the formation of double inversion layers during the whole haze event. The formation of the upper–level inversion layer at around 1200–1600 m should be closely associated with the advection of southwest warm and humid air. Temperature, RH and wind distribution in 925 and 850 hPa at 08:00 BST on 13 April 2014 are shown in Fig.11, from which we could see the weak warm advection from the southwest and west at 925 hPa and 850 hPa, respectively. Relatively high RH values were also observed at the height between 0.7 km and 1.6 km in Fig.7 (b). The nighttime low–level inversion layer below 150 m should be formed by surface longwave radiation cooling while the formation of the daytime low–level inversion layer at around 150–600 m was complex and hard to explain with simple factors. The temperature profiles in the daytime in Fig.10 (a) show that a shallow and weak inversion layer was initially formed at 08:00 on 12 April. And then a deep inversion layer slightly above the surface was formed on 13 April and further developed on 14 April. The air cooling at the low–level inversion top where aerosols accumulated was obvious from 13–14 April. However, the formation of a deep inversion layer from 150–600 m was hard to explain by cooling process of aerosol loadings. It is obvious that with the increased accumulation of polluted aerosols in the daytime the low–level stable layer height was increased and its stability was further strengthened. For example, the inversion layer from 150 m to 550 m with the lapse rate of air temperature is –0.38 ℃ (100 m)$^{-1}$ at 8:00 on 13 April, while the lapse rate of the same layer is –0.75 ℃ (100 m)$^{-1}$ at the same time on 14 April.

The formation of the deep inversion layer in the daytime has not been well understood. Some researchers suggested that the heating process due to the solar radiation absorption by aerosols such as black carbon might play an important role in forming the deep inversion layer (Ding et al., 2016; Petäjä et al., 2016). This might be true for the formation of the deep low–level inversion layer shown in Fig.10, but it is hard to explain the temperature increase in the whole vertical boundary layer and downward movement of air shown in Fig.9. It seems that the descending of the upper–level inversion layer was critical to increase the temperature and humidity in the whole boundary layer in the daytime based on the changes of their profiles in Fig.10.

In all, the persistent advection of southwest warm and humid air provided a long–lasting favourable condition for the formation of a stable upper–level inversion layer of PBL that weakened the upward mixing and diffusing of surface polluted aerosols. As long as the accumulation of aerosols reached certain concentration, the surface cooling induced by the high aerosol loading via blocking the incoming solar radiation became obvious and formed a low–level inversion layer. The formation of the low–



level inversion and subsequent further cooling induced a descending process of PBL height and upper–level inversion layer. The descended warm and humid air from the upper inversion layer significantly strengthen the low–level stability and in return rapidly increase the aerosol loadings. The positive feedback was particularly strong when the $PM_{2.5}$ mass concentration was larger than 150–200 µg m$^{-3}$. Therefore, the descended upper–level inversion layer should be an important factor in strengthening the stability in the whole PBL.

### 3.4.2 Typical fog–haze mixed event

The typical fog–haze mixed event started at 22:00 on 6 October and ended at 18:00 on 11 October 2014 with duration of 117 h. During the whole period, the North China Plain was controlled by the westerly airflow in the mid–troposphere. At the surface, a weak pressure field maintained before the arrival of the cold front, with light winds and high humidity. Moreover, the drizzle rain occurred in Beijing in the morning of 8 October, which further increased the atmospheric humidity in the PBL. The radiation fogs formed from 9 to 11 October.

Figure 12 shows the temporal evolution of surface meteorological and environmental factors in the whole process of the fog–haze mixed event. During the fog–haze mixed events, wind speed varied from 0 m s$^{-1}$ to 2.7 m s$^{-1}$, with an average of 0.5 m s$^{-1}$. The wind direction was easterly from the midnight to afternoon and then changed to calm wind until the next day morning. The temperature was in the range of 9.1 ℃ to 21.7 ℃, with an average of 15.6±3.1 ℃, while the RH was in the range of 46 % to 100 %, with an average of 88±14 %. The visibility exponentially decreased with the $PM_{2.5}$ mass concentration increasing with the $R^2$ of 0.87. The visibility decreased to the minimum 534 m in the morning of 11 October. $PM_{2.5}$ reached the highest at 19:00 on 9 October with the value of 392 µg m$^{-3}$, corresponding to the hourly mean visibility of 898 m. $PM_{2.5}$ decreased dramatically after 17:00 on 11 October. The visibility continued to rise until the end of the event. The averaged $PM_{2.5}$/$PM_{10}$ was as high as 0.94.

The temporal variation of vertical distributions of temperature, RH, LWC and vapor density retrieved by MWRP during the whole fog–haze mixed event is shown in Fig.13. When the precipitation events happened in the morning of 8 October, the profiles became unreliable due to contamination of rainwater on the sensor covering. The LWC was larger than 0.02 g m$^{-3}$ in the morning of 10 and 11 October, which indicated the fog formation (Guo and Guo, 2015). Moreover, the RH was high near the surface in the morning from 8 to 11 October. Relatively high RH values were also observed at the height between 0.5 km to 1.6 km. Compared to the radiosonde temperature profiles in Fig.16 (a) and (c), the PMWR could not capture the temperature inversions at the upper level. Compared with the haze event, the fog–haze mixed event had a higher RH and induced the fog and drizzle formation, so the surface cooling induced by blocking the incoming solar radiation and subsequent descending process of PBL height became more obvious.

Figure 14 shows the time–height distribution series of the backscatter density detected by CL31 and the NRB detected by MPL, and time evolution of the MPL–derived PBL height and $PM_{2.5}$ mass concentration during the whole fog–haze mixed event. As seen in the Fig.14 (a) and (b), aerosols were mostly confined to a shallow layer of few hundred meters. Due to the longer lidar wavelength, the CL31 has better detection capability of the raindrop and fog droplets compared to the MPL. Comparing with





the haze event, the fog and rain drops had stronger attenuation to the signal of MPL. So that the period 1 and 2 in the Fig.14 (b) were caused by drizzles, and in the period 3, 4 and 5, the strong attenuation was caused by fog droplets occurred in the high RH conditions. The daily averaged PBL heights from 7 to 11 October were 0.66, 0.35, 0.27, 0.27 and 0.27 km, respectively, while the daily averaged $PM_{2.5}$ concentrations were 122.7, 249.7, 333.3, 310.8, and 235.4 µg m$^{-3}$, respectively, indicating that

the $PM_{2.5}$ concentration increased with the decease of PBL height. Figure 14 (c) showed that the feedback between the $PM_{2.5}$ concentration and PBL height was much stronger in the fog–haze mixed event than that in haze event. It is obvious that the negative feedback between the $PM_{2.5}$ concentration and PBL height was much weak when the $PM_{2.5}$ concentration was less than 200 µg m$^{-3}$, and it became much strong when the $PM_{2.5}$ concentration reached more than 200 µg m$^{-3}$.

The time–pressure distribution of vertical velocities during the fog–haze event in Beijing from 6 to 11 October 2014 is

10 presented in Fig.15. Similar to the haze event in Sect. 3.4.1, the downdraft zone started to form at the upper levels of boundary layer in the afternoon 7 October, and lasted until a cold frontal system passed the area. The downdraft zone decreased from 700 to 850 hPa in the afternoon 7 October, which led to the sharply decrease of PBL height in Fig.14 (c). In 9 October, the height of downdraft zone was the lowest and the updraft speed in the PBL was the smallest, corresponding to the most polluted day during the whole fog–haze mixed event.

The surface decrease of radiation parameters was more obvious in the fog–haze mixed event shown in Table 4. Daily total horizontal (vertical) plane direct radiation was significantly reduced from14.11 (24.94) to 0.86 (1.4) MJ m$^{-2}$ d$^{-1}$ during 6–11 October 2014, while the total scattering radiation was increased from 3.4 to 7.21 MJ m$^{-2}$ d$^{-1}$. These decreases are consistent with the high $PM_{2.5}$ concentration recorded.

Similar to the haze event, the temperature and RH profiles of the fog–haze mixed event also had stronger double inversion

layers (Fig.16). The obvious upper inversion layer was closely associated with strong advection of warm and humid airflow from southwest (Fig.17). Relatively high RH values were also observed at the height between 0.5 km to 1.7 km in Fig.13 (b). The low–level inversion layer from the surface to 300 m in the daytime had the lapse rate of −1.4 ℃ (100 m)$^{-1}$ at 8:00 on 9 October. The daily averaged PBL height reached the minimum of 0.27 km, and lasted until the end of the fog–haze mixed event in the evening 11 October. As discussed above, the stronger daytime low–level inversion in the fog–haze mixed event

should be related to the higher aerosol concentration accumulated in the low–level. The higher aerosol concentration caused an obvious decrease of surface solar radiation and surface temperature, meanwhile, the absorption of light–absorbing particles such as black carbon increases the temperature above the surface. These processes formed the stronger low–level temperature inversion during the fog–haze mixed event and induced a more stable PBL. The previous studies showed that the black carbon could contribute a fraction of about 3–15 % to the total mass concentrations in urban air (Yang et al., 2011; Huang et al., 2014).

A total of 476 fires caused by biomass burning were observed by MODIS of Terra (10:30) and Aqua (13:30) during the fog– haze mixed event in China according to the Ministry of Environment Protection of the People's Republic of China ( http://www.zhb.gov.cn/hjzl/dqhj/jgjsjcbg/). The nighttime low–level inversion layer was relatively weaker in the fog–haze mixed event than that in the haze event. This is because that the surface long–wave radiation cooling may rapidly cause the



formation of fog as long as the air saturation condition is reached. When the fog is formed, the released latent heating in fog formation will heat the air and weaken the inversion structure.

Table 5 summarizes the averaged $PM_{2.5}$ mass concentration, PBL height, RH and radiation parameters in the haze and fog–haze mixed event. The averaged $PM_{2.5}$ concentration was 164.5 and 250.4 μg m$^{-3}$ in the haze event and fog–haze mixed event, respectively. The difference between surface solar radiations of the haze event and fog–haze mixed event was mainly due to the column accumulation of aerosol and fog drops extinction. Compared with the haze event, the reduction of the surface solar total radiation was 8.88 MJ m$^{-2}$ in the fog–haze mixed event. The total horizontal plane direct radiation and vertical plane direct radiation decreased 4.9 and 5.95 MJ m$^{-2}$, respectively. The average of total scattering radiation rate (the radio of total scattering radiation and total radiation) increased by 17 %. The radiation reduction imposed by aerosol particles was particularly stronger during the fog–haze mixed event. Solar radiation absorbed by the surface of the Earth, heats the bottom of atmospheric column producing convective eddies that transport heat and water vapor upward driving the growth of the PBL (Lee and Ngan, 2011). So, the daily averaged PBL height was lower by 0.11 km in the fog–haze mixed event.

## 4 Conclusion and discussions

In this study, the relationship among PBL height, $PM_{2.5}$ mass concentration and visibility for long–lasting haze and fog–haze mixed events in Beijing was investigated and quantified. Comprehensive measurements of aerosol characteristics and meteorological conditions have been conducted in Chinese Academy of Meteorological Sciences (CAMS), Beijing since 2013, and a total of 11 long–lasting haze and fog–haze mixed events were observed from January 2014 to March 2015. PBL heights of haze and fog–haze mixed events were retrieved using MPL NRB signal and well correlated to the PBL height derived by CL31.

The statistical results show that there was a negative exponential function between the visibility and the $PM_{2.5}$ mass concentration for both haze and fog–haze mixed events with the same $R^2$ of 0.80. Aerosols could cause greater visibility impairments in fog–haze mixed events due to the increase of RH and formation of more fog drops.

The $PM_{2.5}$ concentration had inversely linear correlation with PBL height for haze events with the $R^2$ of 0.34 and negative exponential correlation with the $R^2$ of 0.49 for fog–haze mixed events, indicating that the $PM_{2.5}$ concentration is more sensitive to PBL height in fog–haze mixed events. The feedback between PBL height and $PM_{2.5}$ mass concentration became stronger when $PM_{2.5}$ mass concentration was more than 150–200 μg m$^{-3}$, particularly in fog–haze mixed cases, which is similar to the finding in haze events in Nanjing city (Petäjä et al., 2016). However, our investigation shows that this phenomenon became more obvious in  the fog–haze mixed event.

Similarity to the relationship between $PM_{2.5}$ concentration and PBL height, a positive linear correlation with the $R^2$ of 0.35 existed in haze events and positive exponential correlation with the $R^2$ of 0.55 existed in fog–haze mixed events between visibility and PBL height.



Two typical cases representing for haze and fog–haze mixed events are presented and discussed. The main results show the obvious double inversion layers located at upper–level and low–level respectively formed in both cases. The formation of upper–level inversion layer was closely associated with the persistent advection of southwest warm and humid air. The nighttime low–level inversion layer was formed due to the surface longwave radiation cooling, while the formation of daytime

low–level inversion layer was complex. Our study shows that the initial daytime low–level inversion layer should be related to the surface cooling via blocking the incoming solar radiation by high aerosol loadings. The subsequent rapid development of deep low–level inversion layer in the daytime is hard to explain only by the surface cooling process. We suggest that both the heating process due to the solar radiation absorption by accumulated aerosols such as black carbon and the descended upper–level warm and humid air induced by the surface cooling played critical role in the daytime. Since the strong surface

cooling could collapse PBL and cause an obviously lowing PBL height, and this process could also cause the warm and humid air at upper inversion layer to move downward and induced an increase of temperature and humidity at low air levels. The process can be clearly shown in the case of this study and it might be an important factor in strengthening low–level stability in the daytime. The positive feedback was particularly strong when the $PM_{2.5}$ mass concentration was larger than 150–200 μg m$^{-3}$. Therefore, the descended upper–level inversion layer should be an important factor in strengthening the stability in the

whole PBL.

**Competing interests**

The authors declare they have no conflict of interest.

**Acknowledgements.**

This research was supported by the Research and Development Special Fund for Public Welfare Industry (Meteorology)

(GYHY200806001, GYHY201306047 and GYHY201406001), the National Natural Science Foundation of China (41605111), the Chinese Academy of Meteorological Sciences Basic Research and Operation Fund (2016Z004).

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




**Table 1: The long–lasting haze and fog–haze mixed events from January 2014 to March 2015 in Beijing city**

| Type | Starting date / time | Ending date / time | Minimum visibility (m) | Duration (h) | Maximum PM$_{2.5}$ (µg m$^{-3}$) | Weather phenomenon |
|---|---|---|---|---|---|---|
| Haze events | 2014.01.21/ 15:00 | 2014.01.24/ 15:00 | 1364 | 73 | 264 | – |
| | 2014.04.11/ 22:00 | 2014.04.14/ 23:00 | 1113 | 74 | 304 | – |
| | 2015.02.12/ 21:00 | 2015.02.16/ 10:00 | 1667 | 86 | 263 | – |
| | 2015.03.04/ 22:00 | 2015.03.08/ 10:00 | 1886 | 83 | 266 | – |
| Fog–haze mixed events | 2014.02.19/ 21:00 | 2014.02.26/ 20:00 | 647 | 168 | 269 | 02.26/16:00–21:25 Drizzle rain |
| | 2014.03.22/ 22:00 | 2014.03.28/ 14:00 | 664 | 134 | 417 | 3.28/4:30–6:20 Drizzle rain |
| | 2014.10.06/ 22:00 | 2014.10.11/ 18:00 | 500 | 117 | 391 | 10.08/6:40–7:50 Drizzle rain 10.08/10:30–11:50 Drizzle rain |
| | 2014.10.16/ 21:00 | 2014.10.20/ 23:00 | 964 | 99 | 322 | – |
| | 2014.10.22/ 4:00 | 2014.10.26/ 4:00 | 258 | 97 | 379 | – |
| | 2014.10.28/ 23:00 | 2014.11.01/ 6:00 | 837 | 80 | 184 | 10.29/23:00–10.30/00:10 Drizzle rain 10.31/15:10–16:30 Drizzle rain |
| | 2015.01.11/ 18:00 | 2015.01.16/ 3:00 | 526 | 102 | 297 | 01.14/10:00–10:20 snow |





**Table 2: The exponential curve of visibility and PM2.5 mass concentration, compared with other studies**

| Function | $R^2$ | Reference |
|---|---|---|
| y=0.13974+13.20504exp(−0.00902x) | 0.75 | Han et al. (2016) |
| y=0.08757+12.48402exp(−0.00787x) | 0.79563 | This study (haze events) |
| y=1.04397+26.20687exp(−0.0196x) | 0.80668 | This study (fog–haze mixed events) |

x respects the mass concentration of PM2.5 ($\mu g\ m^{-3}$); y respects visibility (km).

**Table 3: Parameters of radiation from Beijing Meteorological station during the haze event**

| Date | 20140412 | 20140413 | 20140414 |
|---|---|---|---|
| Daily total radiation* (MJ $m^{-2}$) | 18.48 | 18.74 | 17.56 |
| Daily total scattering radiation (MJ $m^{-2}$) | 10.86 | 11.07 | 12.65 |
| Daily total horizontal plane direct radiation (MJ $m^{-2}$) | 7.62 | 7.67 | 4.91 |
| Daily total vertical plane direct radiation (MJ $m^{-2}$) | 10.15 | 10.63 | 6.41 |
| Daily max total radiation (W $m^{-2}$) | 835 | 751 | 715 |
| Daily max vertical plane direct radiation (W $m^{-2}$) | 498 | 421 | 268 |

* Daily total radiation is the sum of daily total scattering and daily total horizontal plane direct radiation.



**Table 4: Parameters of radiation from Beijing Meteorological station for the fog–haze mixed event**

| Date | 2014 1006 | 2014 1007 | 2014 1008 | 2014 1009 | 2014 1010 | 2014 1011 |
|---|---|---|---|---|---|---|
| Daily total radiation* (MJ m$^{-2}$) | 17.51 | 12.37 | 9.02 | 8.39 | 9.05 | 8.07 |
| Daily total scattering radiation (MJ m$^{-2}$) | 3.4 | 7.08 | 7.85 | 7.58 | 8.05 | 7.21 |
| Daily total horizontal plane direct radiation (MJ m$^{-2}$) | 14.11 | 5.29 | 1.17 | 0.81 | 1 | 0.86 |
| Daily total vertical plane direct radiation (MJ m$^{-2}$) | 24.94 | 8.28 | 2.23 | 1.82 | 1.84 | 1.4 |
| Daily max total radiation (W m$^{-2}$) | 749 | 551 | 564 | 385 | 434 | 540 |
| Daily max vertical plane direct radiation (W m$^{-2}$) | 893 | 366 | 148 | 91 | 132 | 202 |

* Daily total radiation is the sum of daily total scattering and daily total horizontal plane direct radiation.

**Table 5 The average PM$_{2.5}$ concentration, PBL height, RH and daily radiation parameters during the haze and fog–haze mixed event**

| | Haze event | Fog–haze mixed event |
|---|---|---|
| PM$_{2.5}$ (μg m$^{-3}$) | 164.5 | 250.4 |
| PBL height (km) | 0.47 | 0.36 |
| RH (%) | 55 | 88 |
| Total radiation (MJ m$^{-2}$) | 18.26 | 9.38 |
| Total scattering radiation (MJ m$^{-2}$) | 11.53 | 7.55 |
| Total horizontal plane direct radiation (MJ m$^{-2}$) | 6.73 | 1.83 |
| Total vertical plane direct radiation (MJ m$^{-2}$) | 9.06 | 3.11 |
| Max total radiation (W m$^{-2}$) | 767 | 398 |
| Max vertical plane direct radiation (W m$^{-2}$) | 396 | 188 |





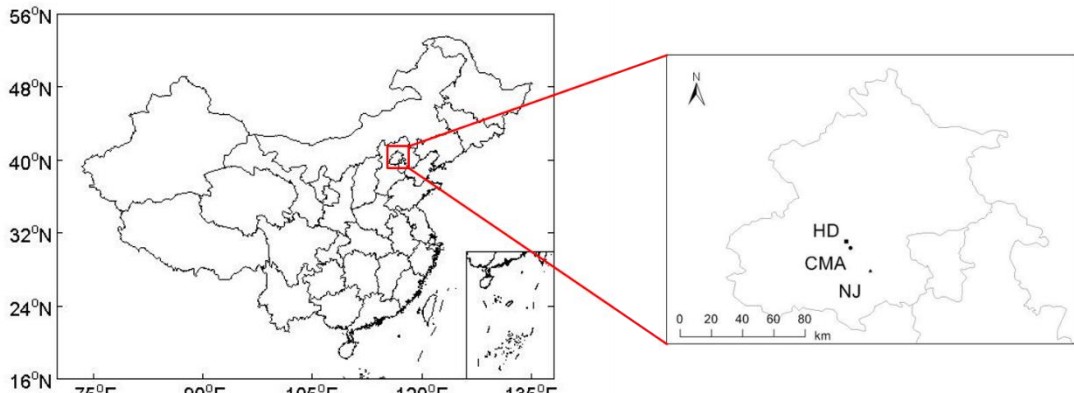

**Figure 1: Geographical location of the observation sit in Beijing. CMA, HD and NJ represent China Meteorological Administration, Haidian automatic weather station and Beijing Meteorological station, respectively.**

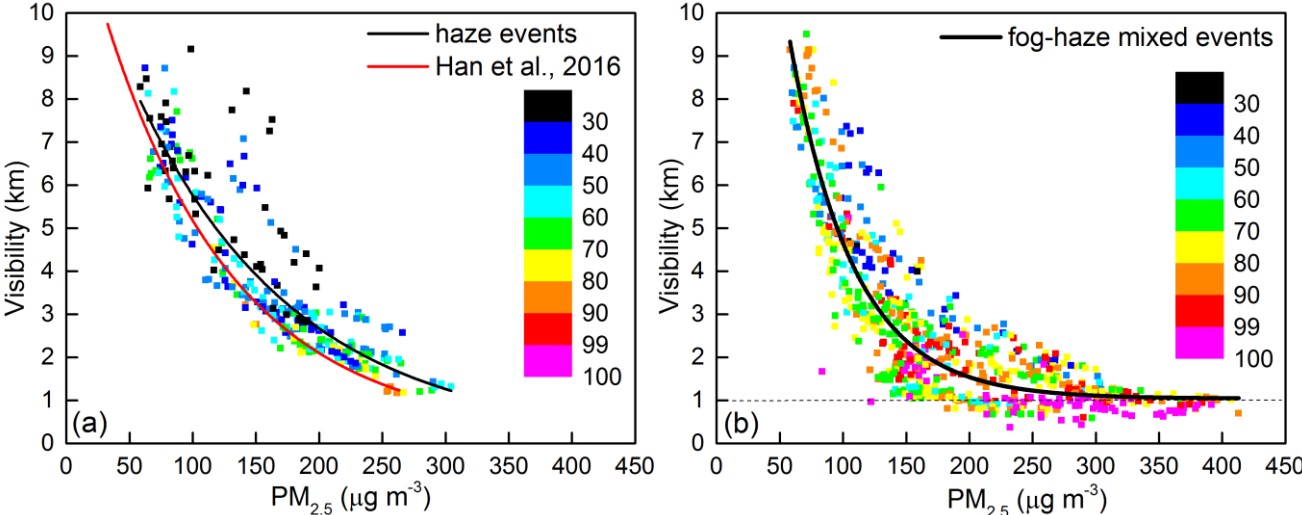

**Figure 2: Relationship between the measured visibility and PM₂.₅ mass concentration under different RH conditions for (a) haze and (b) fog–haze mixed events from January 2014 to March 2015 in Beijing city. The black exponential curves present the fits of the squares. The red exponential curve is the fit of daily averaged visibility and PM₂.₅ concentration from October 2013 to September 2014 on stable meteorological days in Han et al. (2016)**



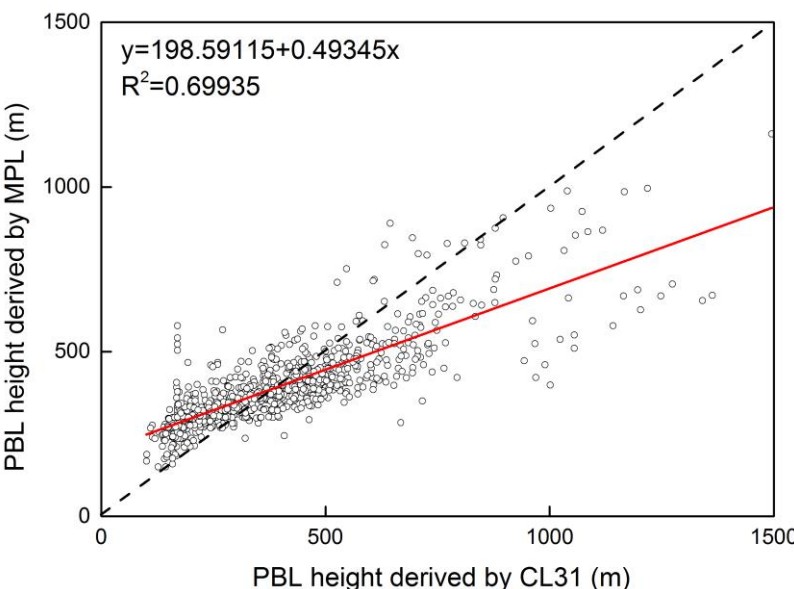

**Figure 3: Comparison of PBL heights derived by MPL and by CL31**

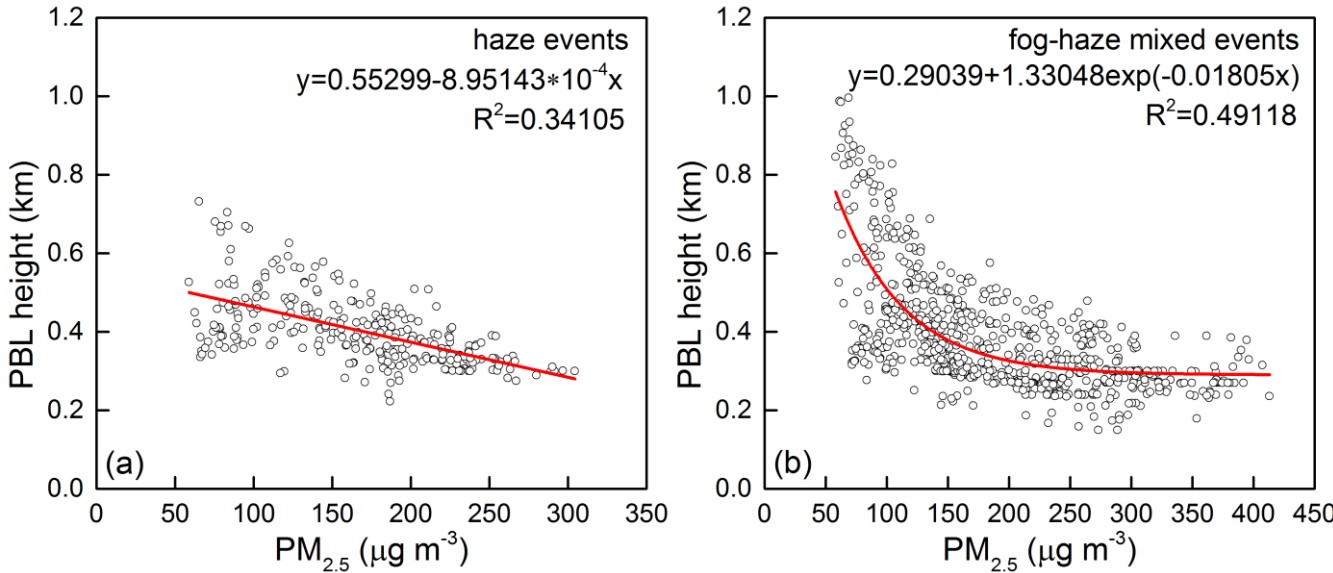

**Figure 4: Relationship between PBL height and PM$_{2.5}$ mass concentration for (a) haze and (b) fog–haze mixed events from January 2014 to March 2015 in Beijing city**





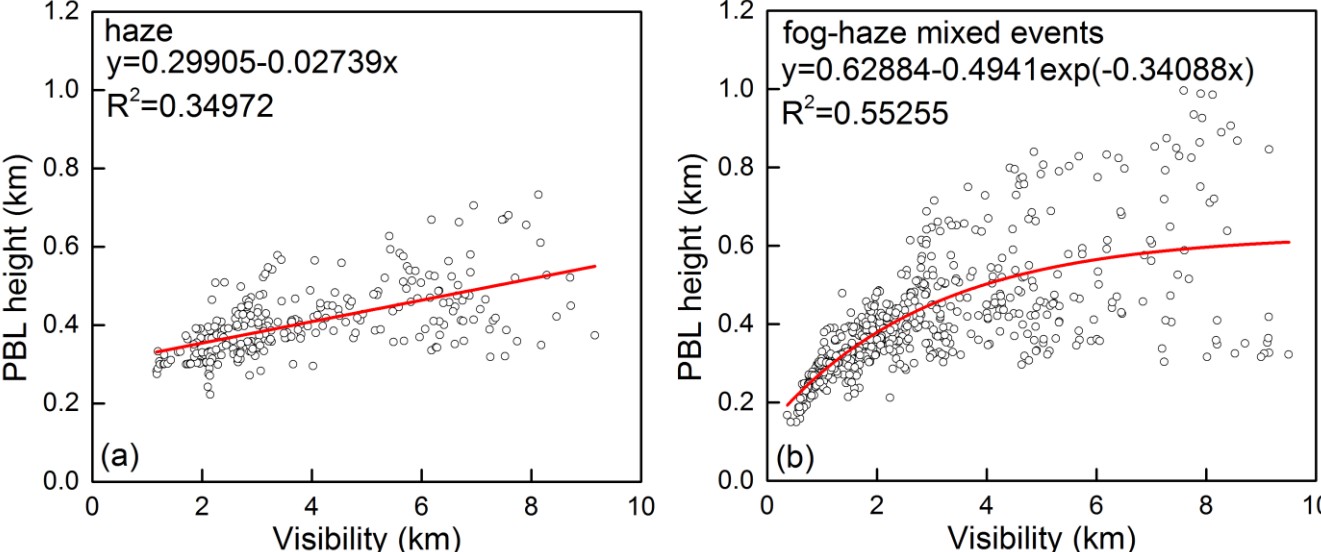

**Figure 5: Relationship between visibility and PBL height for (a) haze and (b) fog-haze mixed events from January 2014 to March 2015 in Beijing city**





**Figure 6: Temporal variations of surface meteorological and environmental parameters observed during the whole haze event in Beijing city. (a) temperature, (b) RH, (c) wind direction and wind speed, (d) visibility, (e) mass concentration of particulate matter. The temperature and RH derived from PMWR is also presented in (a) and (b).**





**Figure 7: Time–height cross sections of (a) temperature, (b) RH, (c) LWC and (d) vapor density retrieved by MWRP during the whole haze event in Beijing city.**





**Figure 8: Time–height cross sections of (a) the backscatter density detected by the CL31 and (b) the NRB detected by the MPL, and (c) the time evolution of PM2.5 mass concentration and PBL height retrieved by the NRB of MPL during the whole haze event in Beijing city.**





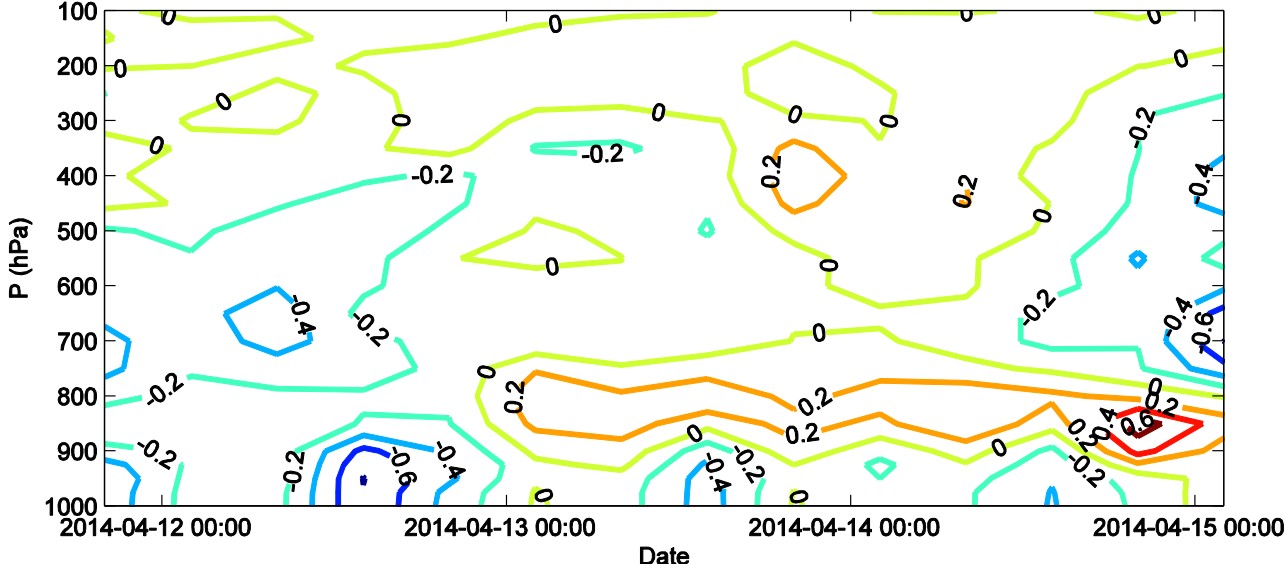

**Figure 9: Time–pressure distribution of vertical velocities during the whole haze event in Beijing city. (Negative and positive numbers stand for the updrafts and downdrafts, respectively. Unit: Pa s$^{-1}$)**





**Figure 10: The temperature and RH profiles during the whole haze event in Beijing city. (a) temperature and (b) RH at 8:00 LST. (c) temperature and (d) RH at 20:00 LST.**





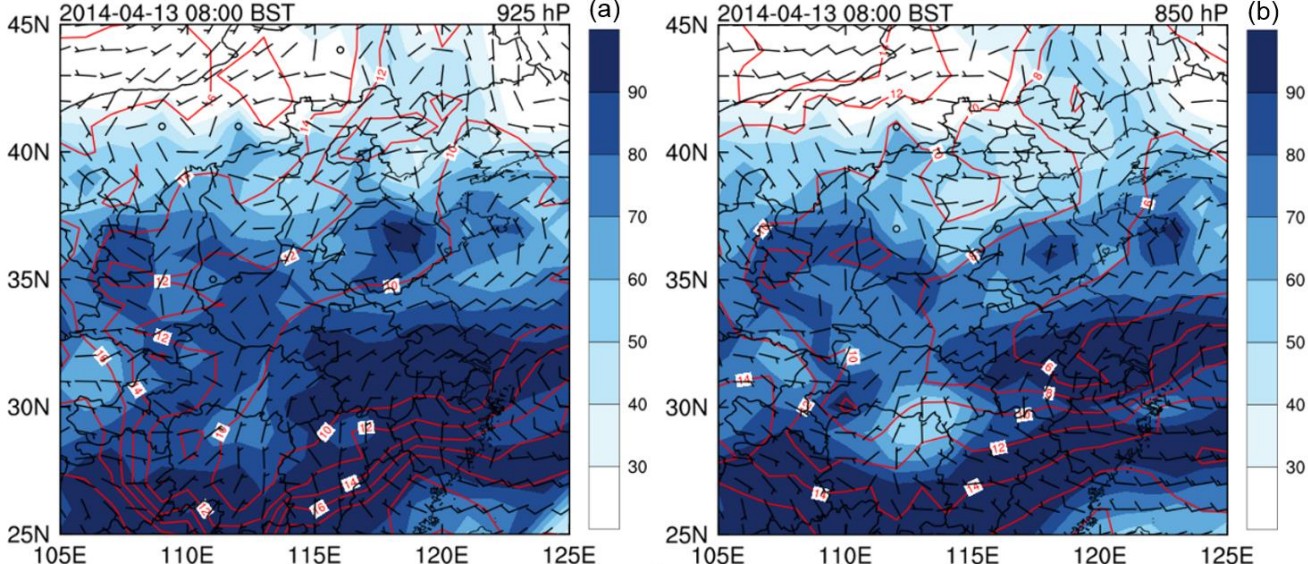

**Figure 11: Temperature (red contour lines, units: ℃), RH (color shading, units: %) and wind (wind bar) distribution in (a) 925 hPa and (b) 850 hPa at 08:00 BST on 13 April 2014**





**Figure 12: Temporal variation of surface meteorological and environmental factors observed during the whole fog–haze mixed event in Beijing city. (a) temperature; (b) RH; (c) wind direction and wind speed; (d) visibility; (e) mass concentration of particulate matter. The temperature and RH derived from PMWR is also presented in (a) and (b).**







**Figure 13: Time–height cross sections of (a) temperature, (b) RH, (c) LWC and (d) vapor density retrieved by MWRP during the whole fog–haze mixed event in Beijing city.**



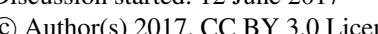



**Figure 14: Time–height cross sections of (a) the backscatter density detected by CL31 and (b) the NRB detected by MPL, and (c) the time evolution of PM2.5 mass concentration and PBL height retrieved by the NRB of MPL during the fog–haze mixed event in Beijing city.**




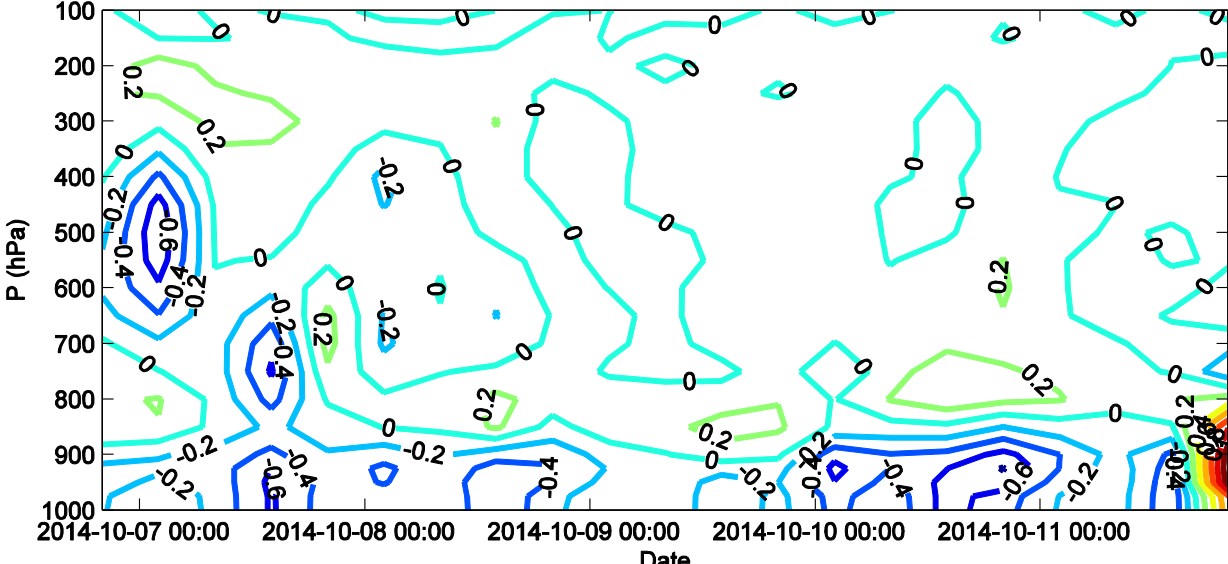

**Figure 15: Time–pressure distribution of vertical velocities during the whole fog–haze mixed event in Beijing city. (Negative and positive numbers stand for the updrafts and downdrafts, respectively. Unit: Pa s⁻¹)**



**Figure 16: The temperature and RH profiles during the whole fog–haze mixed event in Beijing city. (a) temperature and (b) RH at 8:00 LST. (c) temperature and (d) RH at 20:00 LST.**





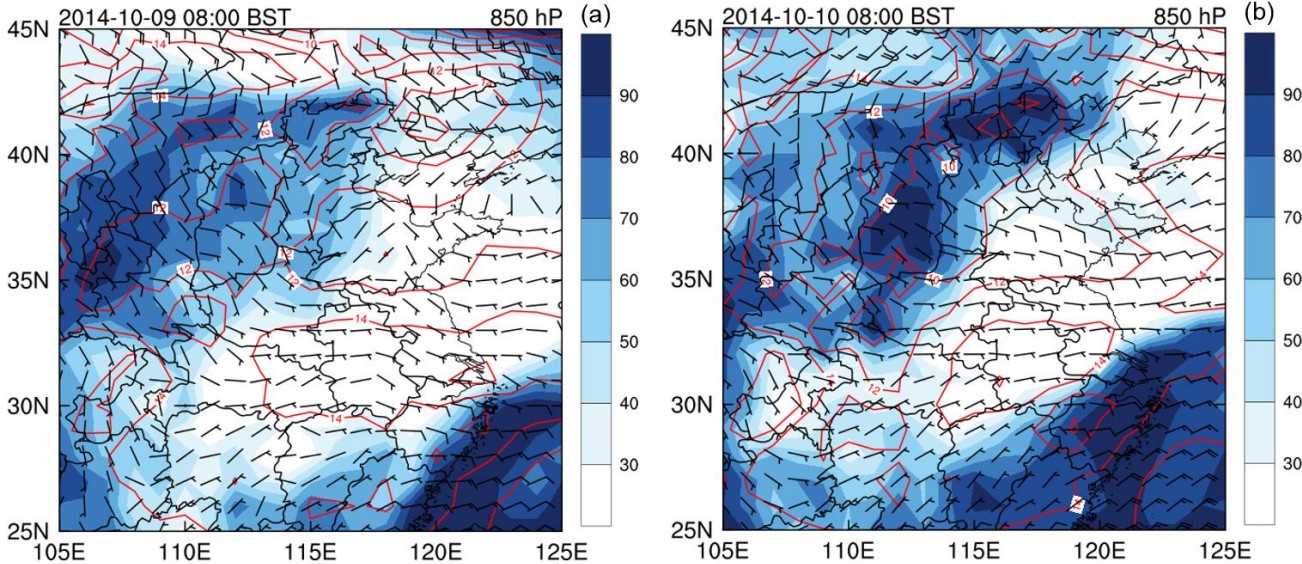

**Figure 17: Temperature (red contour lines, units: ℃), RH (color shading, units: %) and wind (wind bar) distribution in 850 hPa at 08:00 BST on (a) 9 October and (b) 10 October 2014**

