# Peer review of "Quantifying the relationship among PM2.5 concentration, visibility and planetary boundary layer height for long–lasting haze and fog– haze mixed events in Beijing city"

_Atmospheric Chemistry and Physics, 2017_

## Referee Comment (RC1) · Anonymous Referee #3 · 20 Jun 2017

Review of "Quantifying the relationship among PM2.5 concentration, visibility and planetary boundary layer height for long–lasting haze and fog–haze mixed events in Beijing city" by Tian Luan et al. (acp-2017-455)

Summary: In this article, the authors analyzed and quantified the relationships among PM2.5 concentration, visibility and PBL height for the haze and fog-haze mixed events using the data from several state-of-the-art instruments, and then showed the corresponding meteorological conditions for the two typical cases. Similar analyses have been implemented by many previous studies and the novelty of this study is actually

not enough. However, the detailed estimations of this study can still provide some valuable information for the haze early warnings. I suggest it to be accepted after several corrections. Note I am not an expert on the atmospheric chemistry, so my assessment on this part may not be accurate.

Specific comments: 1. For any journal, the first requirement is that the abstract of the article should be briefly and concisely. However, the abstract of this study is too redundant and including some valueless information that would be lowering the readability. So, this part of the article is suggested to be re-worded in the next version that just the highlights from this research are needed.

2. Since 2013, increased studies have addressed the impact of climate changes on the haze pollutions over China. For example, weakened East Asian winter monsoon (Li, Qiang, et al., 2016: Interannual variation of the wintertime fog-haze days across central and eastern China and its relation with East Asian winter monsoon. Int. J. Climatol., 36, 346-354), reduced Arctic sea ice (Wang, Huijun, et al., 2015: Arctic sea ice decline intensified haze pollution in eastern China, Atmos. Oceanic Sci. Lett., 8, 1–9), Tibetan Plateau warming (Xu, X., et al., 2016: Climate modulation of the Tibetan Plateau on haze in China. Atmos. Chem. Phys., 16, 1365-1375), ENSO variability (Gao, Hui, et al., 2015: Influences of El Nino Southern Oscillation events on haze frequency in eastern China during boreal winters. Int. J. Climatol., 35, 2682-2688), etc. all showed important roles on the haze occurrences across China. I think this part of the work should be reviewed in the introduction. Additionally, there are also some studies presented the meteorological conditions for the haze pollutions from climatological perspectives (Zhang, Renhe, et al., 2014: Meteorological conditions for the persistent severe fog and haze event over eastern China in January 2013. Sci. China Earth Sci., 57, 26–35; Chen, Huopo, et al., 2015: Haze days in North China and the associated atmospheric circulations based on daily visibility data from 1960 to 2012. J. Geophys. Res. Atmos., 120, 5895-5909), which can be compared with the case analyses in this study, further increasing the readability.

3. The difference of the separating criterions of the long-lasting haze and fog-haze mixed events is the different value of minimum visibility that minimum visibility larger than 1km for haze events and smaller than 1km for fog-haze mixed events. This is the self-criterion or from the other research? The humidity is a key factor for the separation of the fog and haze events, why the relative humidity has not been considered?

4. In the context, the authors mentioned that "The PBL height derived by MPL is usually used as a reference in detecting the aerosol vertical distribution by more advanced and powerful lidars.", however, the authors also mentioned in the following paragraph that there are also some uncertainty existed for the MPL to determine the PBL height. This seems to be conflict.

5. How about the statistical relationship between PM2.5 ma concentration and PBL height from CL lidars?

6. In this study, the authors just showed the meteorological conditions for two typical haze events? Why chose these two cases from 11 cases? The composite analysis method is suggested for the further analysis if conveniently.

7. The authors presented detailed analyses on the meteorological conditions for the long-lasting haze and fog-haze mixed events. However, I am still not clear what the difference for the meteorological conditions between these two cases. So, the comparison discussion in this aspect should be added in the section of Conclusion and Discussions, not just showing the common features as the current MS did.

8. To increase the readability, the location of Beijing is suggested to be highlighted in Figure 11 and 17.

---

## Referee Comment (RC2) · Anonymous Referee #1 · 26 Jun 2017

This study quantifies the relationship among PM2.5 concentration, visibility and planetary boundary layer height for long–lasting haze and fog– haze mixed events in Beijing city. They found negative relationships between visibility and PM2.5, PM2.5 and PBL height. They also found a double inversion layer formed in both typical events, which played critical roles in maintaining and enhancing the long–lasting polluted events. The topic of this paper is interesting and is suitable for publication in this journal. However, some improvements are needed before publication. Following are the major and specific issues:

Major issues:

The authors provided a large amount information of the relationships between PM2.5, visibility and PBL height. However, these relationships are reported in many previous studies. The new finding in this study about influence double inversion layer on the meteorology and PM2.5 needs more attention and discussion.

Specific issues:

Abstract: Causal relationship about 'The air quality and visibility are strongly influenced by aerosol loading and meteorological conditions.' It would be better to revise as 'influenced by aerosol loading, which is driven by meteorological conditions'.

Introduction: Haze in China is a very hot topic and raises a bunch of new studies recently. The authors may cite more recent papers to strengthen this part. Climate change (Cai et al., 2017, Nature Climate), Arctic sea ice loss (Zou et al., 2017, Science Advances) and decadal weakening of winds (Yang et al., 2016, JGR) suggested causes in climate view. Dust-wind interaction (Yang et al., 2017a, Nature Communications) and upwind transport (Yang et al., 2017b, ACP) can also intensify haze in China.

Page 6 Line 25: Why haze and fog-haze events were defined like this? The author mentioned humidity in the introduction but the fog-haze was not defined based on humidity.

Figure 3: Are the both MPL and CL31 at site in Beijing?

Page 8 Line 21: How about these PBL height in haze events?

Page 9 Section 3.4.1: Why the authors chose April 2014 as the typical haze? Haze in northern China are more severe in winter season. How about the results for other haze events identified in this study?

Page 9 Line 25: The unit of visibility is m here but km in previous figures. Please unify units for the whole figures and manuscript.

Figure 6: I am confused that why PMcoarse did not increase with time. If aerosols are accumulated in the boundary layer due to decrease in PBL, all coarse and fine aerosol concentrations are expected to increase.

Page 12 Line 6: Why the authors did not choose the same spring season as the haze event analysis above?

Technique issue:

Too much figures in the manuscript, the authors may move some into supplement.

References: Cai W., K. Li, H. Liao, H. Wang, and L. Wu (2017), Weather conditions conducive to Beijing severe haze more frequent under climate change, Nat. Clim. Change, doi:10.1038/nclimate3249. Y. Zou, Y. Wang, Y. Zhang, J.H. Koo, Arctic sea ice, Eurasia snow, and extreme winter haze in China, Sci. Adv., 3 (2017), p. e1602751 Yang Y. et al. (2017a), Dust-wind interactions can intensify aerosol pollution over eastern China, Nature Communications, 8, 15333. Yang, Y., Wang, H., Smith, S. J., Ma, P.-L., and Rasch, P. J.: Source attribution of black carbon and its direct radiative forcing in China, Atmos. Chem. Phys., 17, 4319-4336, doi:10.5194/acp-17-4319-2017, 2017b. Yang, Y., H. Liao, and S. Lou (2016), Increase in winter haze over eastern China in recent decades: Roles of varia- tions in meteorological parameters and anthropogenic emissions, J. Geophys. Res. Atmos., 121, 13,050–13,065, doi:10.1002/2016JD025136.

---

## Referee Comment (RC3) · Anonymous Referee #2 · 4 Jul 2017

Comments on Quantifying the relationship among PM2.5 concentration, visibility and planetary boundary layer height for long-lasting haze and fog-haze mixed events in Beijing city by T Luan et al.

In this manuscript author reported the observation phenomena describing the tight relationship between PM and PBL in two type haze events occurred in Beijing. Facts are always important for our better understating of the severe haze events. Authors suggest possible feedbacks among PM, PBL, and/or humidity, whereas the PBL play dominant roles. However, in the present version of the manuscript, authors just demonstrate their

co-changes by correlations. The physical explanation why/how the PBL changes need careful and substantial analysis/evidence.

Page 3, line 3, '. . .inside of the surface', misleading.

Page 4, line 26, 'NCEPT', typo

Sections 3.1 and 3.2, as authors indicated the humidity is a very important factor modulating both the PM concentration and visibility, the relationship obtained in Section 3.1 and 3.2 would be biased by humidity. Particularly, to what extent the humidity 'contaminate' the PM2.5-PBL and PM2.5-visibility relationship should be clarified. It seems reasonable to perform additional analysis using data under similar humidity conditions. Otherwise the explanation would be vague.

Page 9, line 14, 'col', typo

Page 9, line 23, As you demonstrated, the temperature advection is important, but why the aerosol transport from the south is weak?

On the PBL changes, what's the role played by the background synoptic processes? Is there any non-aerosol related dynamical/thermal causes in lower troposphere?

Page 12, line 1, inconvincing. It seems actually the whole layer get warmer on 13-14 April. This might help set a higher PBL.

Section 3.4.2, The PBL feedback is much stronger in fog-haze events than in haze event. This conclusion cannot be obtained, until you have ruled out the influence of synoptic processes on the PBL in these anaylzed events.

Page 32, The RH in Figure 10 seems not consistent with Fig.7. For example, high RH above 500m are event from 12 April to 14 April in Figure 7. But in Figure 10 (b,d) this feature cannot be found, instead, much drier conditions on 13 April and 14 April. Why?

Page 34, Figure 12(e), PMcorse should be PMcoarse. And how did you define PMcoarse? Should be indicated, and the relevant analysis for PMcoarse also missed in

the text.

---

## Author Response (AR1)

**Manuscript # acp-2017-455**

**Reply to Referee #1**

We are very grateful to all important and helpful comments from the referee. The followings are our responses to each comment in detail.

This study quantifies the relationship among $PM_{2.5}$ concentration, visibility and planetary boundary layer height for long–lasting haze and fog–haze mixed events in Beijing city. They found negative relationships between visibility and $PM_{2.5}$, $PM_{2.5}$ and PBL height. They also found a double inversion layer formed in both typical events, which played critical roles in maintaining and enhancing the long–lasting polluted events. The topic of this paper is interesting and is suitable for publication in this journal. However, some improvements are needed before publication. Following are the major and specific issues:

Major issues:

1. The authors provided a large amount information of the relationships between $PM_{2.5}$, visibility and PBL height. However, these relationships are reported in many previous studies. The new finding in this study about influence double inversion layer on the meteorology and $PM_{2.5}$ needs more attention and discussion.

**Reply 1:**

Thanks for your constructive suggestions. We added some discussions relevant to our new finding about influence of double inversion layer on the polluted events in the conclusions and discussions section, as "The new finding in this paper has important implications in explaining the frequent long–lasting polluted events in the study region. Generally, a typical pollution event is usually formed under a stable and shallow temperature–inversion condition at low atmospheric layers, and would disappear or obviously decrease when the daytime solar radiation increases. However, in the study region, we found that many severe haze and fog–haze mixed events lasted for several days even for several weeks. Most previous publications attributed the reason as the persistent abnormal weather system or high emissions. However, this study shows that except for the influence of meteorological condition and high emission, the interactions and feedbacks between PBL and aerosol loading linked by radiation process are crucial in enhancing and maintaining these polluted events. These feedbacks could cause an important variation of dynamical/thermal processes in lower troposphere. The formation of double inversion layer and their subsequent change is closely associated with persistent meteorological condition, high aerosol loading and associated radiation process. Due to the complex interactions and feedbacks, a deeper and more stable atmospheric low–level inversion layer is formed and it is hard to break up by daytime solar radiation heating process until the strong wind occurs and removes the high aerosol loading."

Specific issues:

2. Abstract: Causal relationship about 'The air quality and visibility are strongly influenced by aerosol loading and meteorological conditions.' It would be better to revise as 'influenced by aerosol loading, which is driven by meteorological conditions'.

**Reply 2:**

Thanks for the suggestion. It has been corrected in the revised manuscript.

3. Introduction: Haze in China is a very hot topic and raises a bunch of new studies recently. The authors may cite more recent papers to strengthen this part. Climate change (Cai et al., 2017, Nature Climate), Arctic sea ice loss (Zou et al., 2017, Science Advances) and decadal weakening of winds (Yang et al., 2016, JGR) suggested causes in climate view. Dust-wind interaction (Yang et al., 2017a, Nature Communications) and upwind transport (Yang et al., 2017b, ACP) can also intensify haze in China.

**Reply 3:**

Thanks for the good suggestion. We have added some recent papers in the introduction section based on your suggestion, as "In addition, the interactions between aerosol pollution and climate change have been substantially addressed in recent publications, for example, anthropogenic climate change (Cai et al., 2017), reduced Arctic sea ice (Wang et al., 2015; Zou et al., 2017), the Tibetan Plateau warming (Xu et al., 2016), influences of ENSO events on haze frequency in eastern China (Gao and Li, 2015), weakened East Asian winter monsoon (Li et al., 2016), decadal weakening of winds (Yang et al., 2016), and enhanced thermal stability of the lower atmosphere (Zhang et al., 2014; Chen and Wang, 2015). The dust–wind interaction (Yang et al., 2017a) and upwind transport (Yang et al., 2017b) could also intensify haze events in China."

4. Page 6 Line 25: Why haze and fog-haze events were defined like this? The author mentioned humidity in the introduction but the fog-haze was not defined based on humidity.

**Reply 4:**

Thanks for the comment. Actually, the definition we used is from international definition of fog event. Fog is an observed horizontal visibility below 1000 m in the presence of suspended water droplets and/or ice crystals (NOAA, 1995), which means that when the horizontal visibility is below 1000 m, the fog events occur. Since the horizontal visibility for atmospheric haze event is usually larger than 1000 m, only the fog occurs the visibility can decrease to be less than 1000 m. So that is why we use the minimum visibility to define fog and haze events.

Theoretically, when a fog event occurs, the RH has to reach over 100 %. However, it is difficult to measure RH accurately, so in most cases, we use RH value of 90 % or 95 % as criterion to separate fog and haze. In fact, in the study region, when the RH is high enough, the fog and haze are usually co-existed. The haze aerosols can be transformed to fog droplets under certain conditions according to the Köhler curve (Köhler, 1936). It should be noticed that the situations such as heavy rain event or light fog events, which cause the horizontal visibility to be below or above 1000 m are not considered here. In addition, since we focus on long–lasting severe fog and haze event, we also include factors such as the lasting time and $PM_{2.5}$ mass concentration as additional criteria. To be more clearly, the corresponding text have been modified in the manuscript.

We also revised Table 1 to include more parameters such as duration and maximum RH of the pollutant events, show as below:

**Table 1: The long–lasting haze and fog–haze mixed events from January 2014 to March 2015 in Beijing city**

| Type | Starting date / time | Ending date / time | Minimum visibility (m) | Duration (h) | Maximum PM$_{2.5}$ ($\mu$g m$^{-3}$) | Maximum RH (%) | Weather phenomenon |
|---|---|---|---|---|---|---|---|
| Haze events | 2014.01.21/ 15:00 | 2014.01.24/ 15:00 | 1364 | 73 | 264 | 68 | – |
| | 2014.04.11/ 22:00 | 2014.04.14/ 23:00 | 1113 | 74 | 304 | 89 | – |
| | 2015.02.12/ 21:00 | 2015.02.16/ 10:00 | 1667 | 86 | 263 | 77 | – |
| | 2015.03.04/ 22:00 | 2015.03.08/ 10:00 | 1886 | 83 | 266 | 72 | – |
| Fog–haze mixed events | 2014.02.19/ 21:00 | 2014.02.26/ 20:00 | 647 | 168/76[b] | 269 | 92 | 02.26/16:00–21:25 Drizzle rain |
| | 2014.03.22/ 22:00 | 2014.03.28/ 14:00 | 664 | 137/13[b] | 417 | 94 | 3.28/4:30–6:20 Drizzle rain |
| | 2014.10.06/ 22:00 | 2014.10.11/ 18:00 | 500 | 117/48[b] | 391 | 100 | 10.08/6:40–7:50 10.08/10:30–11:50 Drizzle rain |
| | 2014.10.16/ 21:00 | 2014.10.20/ 23:00 | 964 | 99/3[b] | 322 | 100 | – |
| | 2014.10.22/ 4:00 | 2014.10.26/ 4:00 | 258 | 97/24[b] | 379 | 100 | – |
| | 2014.10.28/ 23:00 | 2014.11.01/ 5:00 | 837 | 79/1[b] | 184 | 100 | 10.29/23:00– 10.30/00:10 10.31/15:10–16:30 Drizzle rain |
| | 2015.01.12/ 17:00 | 2015.01.16/ 3:00 | 526 | 83/8[b] | 297 | 93[a] | 01.14/10:00–10:20 snow |

[a] the maximum RH of all valid data except missing measurements.
5    [b] fog–haze mixed event duration / fog duration.

5. Figure 3: Are the both MPL and CL31 at site in Beijing?

**Reply 5:**

Yes, they are all located in the observational site at the campus of China Meteorological Administration (CMA) in Beijing. CL31 was sited on the roof of a 20 m tall building, and MPL installed in a working container beside the building not far away 10 m. To be more clearly, we have modified the corresponding text in section 2.1 as "The vertical profiles of aerosol in the troposphere and the PBL height could be also obtained from a ground–based MPL installed in a working container 10 m far away from the building at the campus of CMA."

6. Page 8 Line 21: How about these PBL height in haze events?

**Reply 6:**

Thank you for your questions. The $PM_{2.5}$ concentrations of 100 µg m$^{-3}$, 200 µg m$^{-3}$, 300 µg m$^{-3}$ corresponded to the PBL heights of 460 m, 370 m and 280 m, respectively. It has been added in section 3.2 of the revised manuscript.

7. Page 9 Section 3.4.1: Why the authors chose April 2014 as the typical haze? Haze in northern China are more severe in winter season. How about the results for other haze events identified in this study?

**Reply 7:**

Thank you for your questions. As you said, the haze events in winter season in northern China are usually more severe. As seen in Table 1, there were only four long–lasting haze events during January 2014 to March 2015 in Beijing. The maximum $PM_{2.5}$ concentration was the highest and the corresponding data was relatively complete in the haze event in April 2014, so we choose the case as the type haze event. We have added these descriptions in section 3.4 of the revised manuscript. Our main conclusion about the influence of double inversion layer on the long–lasting pollution events in Beijing city are also suitable to the cases in Table 1.

8. Page 9 Line 25: The unit of visibility is m here but km in previous figures. Please unify units for the whole figures and manuscript.

**Reply 8:**

Thanks for the suggestion. The units of visibility and PBL height as 'm' for the whole figures and texts are unified in the revised manuscript, shown as below:

[Figure]

**Figure 2: Relationship between the measured visibility and PM$_{2.5}$ mass concentration under different RH conditions for (a) haze and (b) fog–haze mixed events from January 2014 to March 2015 in Beijing city. The black exponential curves present the fits of the squares. The red exponential curve is the fit of daily averaged visibility and PM$_{2.5}$ concentration from October 2013 to September 2014 on stable meteorological days in Han et al. (2016)**

[Figure]

**Figure 3: Relationship between PBL height and PM$_{2.5}$ mass concentration for (a) haze and (b) fog–haze mixed events from January 2014 to March 2015 in Beijing city**

[Figure]

**Figure 4: Relationship between visibility and PBL height for (a) haze and (b) fog-haze mixed events from January 2014 to March 2015 in Beijing city**

9. Figure 6: I am confused that why PMcoarse did not increase with time. If aerosols are accumulated in the boundary layer due to decrease in PBL, all coarse and fine aerosol concentrations are expected to increase.

**Reply 9:**

Thanks for the comment. This is caused by the much lower value of PMcoarse than fine aerosol concentration. If we redraw the figures for the haze event and fog–haze mixed event (see below Fig. A and Fig. B), we can find almost the same tendency. The left vertical axis represents the concentration of $PM_{2.5}$ and $PM_{10}$. The right vertical axis represents the PMcoarse ($PM_{2.5-10}$) concentration. In general, the variation trend of PMcoarse is consistent with that of $PM_{2.5}$ in both two cases, especially in the fog–haze mixed event. In the previous manuscript, we used single vertical axis for the concentration of $PM_{2.5}$, PMcoarse and $PM_{10}$ and the variation of PMcoarse concentration is not clear due to its lower value. Since the variation trend of PMcoarse was generally consistent with that of $PM_{2.5}$ in both two cases, so we have deleted the lines of PMcoarse in Fig. 5 (e) and Fig. 10 (e) in the revised manuscript.

[Figure]

**Figure A: Temporal variations of mass concentration of particulate matter observed during the whole haze event in Beijing city.**

[Figure]

5   **Figure B: Temporal variation of mass concentration of particulate matter observed during the whole fog–haze mixed event in Beijing city.**

10. Page 12 Line 6: Why the authors did not choose the same spring season as the haze event analysis above?

**Reply 10:**

10   Thanks for the comment. The fog–haze mixed event from 22 to 28 March, 2014 for the same spring season as the haze event was much weaker and shorter-lived than that we selected (Table 1). The fog in selected fog-haze mixed case lasted about 3 days while that for the case with same season only lasted 13 hours. In order to investigate the typical long–lasting fog–haze mixed event, we chose this case. To be more clearly, we have added the message to section 3.4 of the revised manuscript as

"In all haze events, the haze event observed from 11 to 14 April was highly polluted with the maximum $PM_{2.5}$ concentration of 304 μg m$^{-3}$ and minimum visibility of 1113 m. For all fog–haze mixed events, the fog duration was considered firstly. Two cases are chosen, in which the fog duration accounted for more than 40 % of the total. One was observed from 19 to 26 February 2014, and the other was occurred from 6 to 11 October 2014. Moreover, the maximum RH reached to 100 % in the fog–haze event occurred from 6 to 11 October 2014, which was chosen as typical fog–haze event for the following study."

Technique issue:

11. Too much figures in the manuscript, the authors may move some into supplement.

**Reply 11:**

Thanks for the suggestion. We have moved some figures into supplement in the revised manuscript (see Fig. S1, Fig. S2, Fig. S3 and Fig. S4 in the Supplement).

10  PBL changes need careful and substantial analysis/evidence.

**Reply 1:**

Thanks for the suggestion. We added more explanations and discussions relevant to the PBL changes and PM concentration in conclusions and discussions section of the revised manuscript, as "The new finding in this paper has important implications in explaining the frequent long–lasting polluted events in the study region. Generally, a typical pollution event is usually

15  formed under a stable and shallow temperature–inversion condition at low atmospheric layers, and would disappear or obviously decrease when the daytime solar radiation increases. However, in the study region, we found that many severe haze and fog–haze mixed events lasted for several days even for several weeks. Most previous publications attributed the reason as the persistent abnormal weather system or high emissions. However, this study shows that except for the influence of meteorological condition and high emission, the interactions and feedbacks between PBL and aerosol loading linked by

20  radiation process are crucial in enhancing and maintaining these polluted events. These feedbacks could cause an important variation of dynamical/thermal processes in lower troposphere. The formation of double inversion layer and their subsequent change is closely associated with persistent meteorological condition, high aerosol loading and associated radiation process. Due to the complex interactions and feedbacks, a deeper and more stable atmospheric low–level inversion layer is formed and it is hard to break up by daytime solar radiation heating process until the strong wind occurs and removes the high aerosol

25  loading."

2. Page 3, line 3, '…inside of the surface', misleading.

**Reply 2:**

Thanks for the comment. It has been changed in the revised manuscript, as "The aerosols directly emitted from polluted source

30  and those secondly formed might be concentrated in the PBL, resulting in high concentrations near the surface."

3. Page 4, line 26, 'NCEPT', typo

**Reply 3:**

It has been revised as "NCEP".

4. Sections 3.1 and 3.2, as authors indicated the humidity is a very important factor modulating both the PM concentration and visibility, the relationship obtained in Section 3.1 and 3.2 would be biased by humidity. Particularly, to what extent the humidity 'contaminate' the $PM_{2.5}$-PBL and $PM_{2.5}$-visibility relationship should be clarified. It seems reasonable to perform additional analysis using data under similar humidity conditions. Otherwise the explanation would be vague.

**Reply 4:**

Thanks for the suggestion. We have performed additional analysis using data under similar humidity condition. The relationship between visibility and $PM_{2.5}$ mass concentrations and that between $PM_{2.5}$ mass concentration and PBL height under different RH conditions for both long–lasting haze and fog–haze mixed events are shown in Fig. A and Fig. B. The results show that the variation of RH has some influences on the relationship between $PM_{2.5}$–PBL and $PM_{2.5}$–visibility. In general, the high RH could decrease the visibility and PBL height quicker than the low RH. However, the tendency and basic conclusion are not obviously changed.

[Figure]

**Figure A: Relationship between the measured visibility and $PM_{2.5}$ mass concentration under different RH conditions for (a) haze and (b) fog–haze mixed events from January 2014 to March 2015 in Beijing city. The exponential curves present the fits of the circles according RH.**

[Figure]

**Figure B: Relationship between PBL height and PM$_{2.5}$ mass concentration under different RH conditions for (a) haze and (b) fog–haze mixed events from January 2014 to March 2015 in Beijing city. The curves present the fits of the circles according RH.**

5. Page 9, line 14, 'col', typo

**Reply 5:**

The col pressure filed presents the pressure field of the saddle type. We rewrote the sentence in the revised manuscript as "The synoptic situation during the haze event characterized as a saddle field. Beijing was located in a saddle between two pairs of high and low pressure center."

6. Page 9, line 23, As you demonstrated, the temperature advection is important, but why the aerosol transport from the south is weak?

**Reply 6:**

Thanks for the comment. The sentence in the manuscript was revised as "The wind speed varied from 0 m s$^{-1}$ to 3.9 m s$^{-1}$, with an average of 0.8 m s$^{-1}$, suggesting that the horizontal diffusion of aerosols was very weak."

7. On the PBL changes, what's the role played by the background synoptic processes? Is there any non-aerosol related dynamical/thermal causes in lower troposphere?

Reply 7:

Thank you for your comment. The background synoptic processes played an important role in the PBL changes in the cases we investigated. The upper inversion layer was formed by the persistent warm and humid airflow from south, and the PBL change was directly related to the descending process of the upper inversion layer. In the daytime, due to the existence of the

upper inversion layer, the PBL tends to become stable and aerosol loading increases. As long as the aerosol loading reach certain high value such as that larger than 150–200 μg m$^{-3}$, the solar radiation will be strongly blocked, and then the strong surface cooling occurs, which cause the descending of the upper layer inversion. Since the upper inversion layer contains warmer and humid air, the descending process would cause the whole PBL to be suppressed and well–mixed. These processes will finally form a deeper and more stable PBL, so the daytime radiation heating cannot break up the stable layer and cause a long–lasting pollution event until strong wind comes. We can see that the interaction and feedback between PBL and aerosol loading is linked by radiation process. We cannot find additional influence from synoptic process such as downdraft etc. Moreover, the descending process of upper inversion layer also bring more water vapour to the lower layer, high content moisture might also play an important role in blocking solar radiation except for the aerosol loading.

8. Page 12, line 1, inconvincing. It seems actually the whole layer get warmer on 13-14 April. This might help set a higher PBL.

**Reply 8:**

Thanks for the comment. In general, when temperature within PBL becomes higher, the PBL height should increase. However, this study shows that due to the strong cooling at the surface, the whole PBL descending and the warm PBL is caused by the descending upper warm and humid air, which is favourable to the mixing process within the PBL, but cannot force the PBL to extend upward due to the influence of strong surface cooling process caused by the rapidly increased aerosol loading. The surface cooling is higher in 14 than in 13 April 2014. For example, the inversion layer from 150 m to 550 m with the lapse rate of air temperature is –0.38 ℃ (100 m)$^{-1}$ at 8:00 on 13 April, while the lapse rate of the same layer is –0.75 ℃ (100 m)$^{-1}$ at the same time on 14 April. From the direct radiation and scattering radiation parameters in Table 3, we can also see the surface cooling effect is strengthened (the direct radiation is reduced and the scattering radiation is increased from 13 to 14 April).

9. Section 3.4.2, The PBL feedback is much stronger in fog-haze events than in haze event. This conclusion cannot be obtained, until you have ruled out the influence of synoptic processes on the PBL in these analyzed events.

**Reply 9:**

Thanks for the comment. This study shows that the PBL feedback is much stronger in fog–haze mixed events than in haze event. This is because that the fog–haze events include many fog droplets, which can substantially block the solar radiation comparing with aerosol loading in haze events in the daytime and cause stronger surface cooling. The stronger surface cooling would cause stronger descending of the upper inversion layer and then form a highly suppressed and more stable PBL. So the PBL feedback is much stronger in fog–haze event. As seen in Table 5 in the manuscript, the radiation reduction imposed by aerosol particles is particularly stronger during the fog–haze mixed event than the haze event. The PM$_{2.5}$ concentration is

higher and the PBL heights are lower in the fog–haze mixed event. We propose that the PBL feedback is much stronger in fog–haze mixed events than in haze event. We have added these descriptions in section 3.4.2 of the revised manuscript.

10. Page 32, The RH in Figure 10 seems not consistent with Fig.7. For example, high RH above 500m are event from 12 April to 14 April in Figure 7. But in Figure 10 (b,d) this feature cannot be found, instead, much drier conditions on 13 April and 14 April. Why?

**Reply 10**:

Thanks for the comment. This is primarily caused by different measuring system. Comparing with data from two methods, The RH derived from PMWR has larger uncertainties and the sounding data are more reliable. Since Fig.7 in the previous manuscript (Fig.6 in the revised manuscript) is from PMWR (Profiling microwave radiometer), which uses passive remote sensing way to obtain profiles of temperature and water vapor based on neural network algorithm with the input of past sounding data and bright temperature. So the RH values of PMWR are derived from the PMWR–retrieved temperature and water vapor density. The vertical resolution of PMWR is only 100 m for the height between 500 and 2000 m, which is not enough to obtain the fine structure of the upper boundary layer. However, the RH from PMWR has higher temporal resolution, it is still very useful compared with conventional radiosonde observation, which only has observation twice a day. In the study of (Xu et al., 2015), atmospheric profiles of RH retrieved from PMWR measurements are compared with radiosonde soundings. The correlation coefficients of RH for clear and cloudy skies are less than 0.8 and decrease monotonically with height. The biases increase from ~3 % at the surface to ~15–20 % at 4000–5000 m. It is well known that radiosonde humidity has systematic dry bias relative to the Cryogenic Frostpoint Hygrometer (Vomel et al., 2007; Bian et al., 2011).

11. Page 34, Figure 12(e), PMcorse should be PMcoarse. And how did you define PMcoarse? Should be indicated, and the relevant analysis for PMcoarse also missed in Interactive comment on Atmos. Chem. Phys. Discuss., https://doi.org/10.5194/acp-2017-455,2017.

**Reply 11:**

Thanks for the careful review. In the section 2.1, PMcoarse is defined as that the particulate matter with aerodynamic diameter is >2.5 μm and <=10 μm. Since the averaged $PM_{2.5}/PM_{10}$ in the haze event and fog–haze mixed event was 0.82 and 0.94, respectively. So the primary pollutant was $PM_{2.5}$ in the study region. In general, the variation trend of PMcoarse was consistent with that of $PM_{2.5}$ in both two cases. According to the above reasons, we have deleted the lines of PMcoarse in Figure 5 (e) and Figure 10 (e) in the revised manuscript.

15 the article is suggested to be re-worded in the next version that just the highlights from this research are needed.

**Reply 1:**

Thanks for the suggestion, the abstract has been re-worded in the revised manuscript, shown as below:

"The air quality and visibility are strongly influenced by aerosol loading, which is driven by meteorological conditions. The quantification of their relationships is critical to understanding the physical and chemical processes and forecasting of the

20 polluted events. We investigated and quantified the relationship among $PM_{2.5}$ (particulate matter with aerodynamic diameter is 2.5 μm and less) mass concentration, visibility and planetary boundary layer (PBL) height in this study based on the data obtained from four long–lasting haze events and seven long–lasting fog–haze mixed events from January 2014 to March 2015 in Beijing city. The statistical results show that there was a negative exponential function between the visibility and the $PM_{2.5}$ mass concentration for both haze and fog–haze mixed events. However, the fog–haze events caused a more obvious decrease

25 of visibility than that for haze events due to the formation of fog droplets that could induce higher light extinction. The $PM_{2.5}$ concentration had inversely linear correlation with PBL height for haze events and negative exponential correlation for fog– haze mixed events, indicating that the $PM_{2.5}$ concentration is more sensitive to PBL height in fog–haze mixed events. The visibility had positively linear correlation with the PBL height with the $R^2$ of 0.35 in haze events and positive exponential correlation with the $R^2$ of 0.56 in fog–haze mixed events. We also investigated the physical mechanism responsible for these

30 relationships among visibility, $PM_{2.5}$ concentration and PBL height through typical haze and fog–haze mixed event, and found that a double inversion layer formed in both typical events and played critical roles in maintaining and enhancing the long– lasting polluted events. The upper–level stable inversion layer formed by the persistent southwest warm and humid airflow

caused the $PM_{2.5}$ accumulation and subsequent surface cooling as well as the formation of a weak low–level inversion layer. The formation of low–level inversion layer further enhanced the $PM_{2.5}$ accumulation and surface cooling process, and induced a strong descending process of the upper–level inversion layer with warm and humid air, which significantly strengthened the PBL stability and formed a deep stable PBL in the daytime, and in return rapidly increased the $PM_{2.5}$ concentration. This positive feedback was particularly strong when the $PM_{2.5}$ mass concentration was larger than 150–200 μg m$^{-3}$. Therefore, the formation and subsequent descending processes of the upper–level inversion layer should be an important factor in maintaining and strengthening the long–lasting severe polluted events, which has not been revealed in previous publications. The interactions and feedbacks between $PM_{2.5}$ concentration and PBL height linked by radiation process caused an obvious and more rapid increase of $PM_{2.5}$ concentration and a significant and long–lasting deterioration of air quality and visibility in fog–haze mixed events."

2. Since 2013, increased studies have addressed the impact of climate changes on the haze pollutions over China. For example, weakened East Asian winter monsoon (Li, Qiang, et al., 2016: Interannual variation of the wintertime fog-haze days across central and eastern China and its relation with East Asian winter monsoon. Int. J. Climatol., 36, 346-354), reduced Arctic sea ice (Wang, Huijun, et al., 2015: Arctic sea ice decline intensified haze pollution in eastern China, Atmos. Oceanic Sci. Lett.,8, 1–9), Tibetan Plateau warming (Xu, X., et al., 2016: Climate modulation of the Tibetan Plateau on haze in China. Atmos. Chem. Phys., 16, 1365-1375), ENSO variability (Gao, Hui, et al., 2015: Influences of El Nino Southern Oscillation events on haze frequency in eastern China during boreal winters. Int. J. Climatol., 35, 2682-2688), etc. all showed important roles on the haze occurrences across China. I think this part of the work should be reviewed in the introduction. Additionally, there are also some studies presented the meteorological conditions for the haze pollutions from climatological perspectives (Zhang, Renhe, et al., 2014: Meteorological conditions for the persistent severe fog and haze event over eastern China in January 2013. Sci. China Earth Sci., 57, 26–35; Chen, Huopo, et al., 2015: Haze days in North China and the associated atmospheric circulations based on daily visibility data from 1960 to 2012. J. Geophys. Res. Atmos., 120, 5895-5909), which can be compared with the case analyses in this study, further increasing the readability.

**Reply 2:**

Thanks for the important suggestion. We have added some recent papers in the introduction section of the revised manuscript, as "In addition, the interactions between aerosol pollution and climate change have been substantially addressed in recent publications, for example, anthropogenic climate change (Cai et al., 2017), reduced Arctic sea ice (Wang et al., 2015; Zou et al., 2017), the Tibetan Plateau warming (Xu et al., 2016), influences of ENSO events on haze frequency in eastern China (Gao and Li, 2015), weakened East Asian winter monsoon (Li et al., 2016), decadal weakening of winds (Yang et al., 2016), and enhanced thermal stability of the lower atmosphere (Zhang et al., 2014; Chen and Wang, 2015). The dust–wind interaction (Yang et al., 2017a) and upwind transport (Yang et al., 2017b) could also intensify haze events in China."

3. The difference of the separating criterions of the long-lasting haze and fog-haze mixed events is the different value of minimum visibility that minimum visibility larger than 1km for haze events and smaller than 1km for fog-haze mixed events. This is the self-criterion or from the other research? The humidity is a key factor for the separation of the fog and haze events, why the relative humidity has not been considered?

**Reply 3:**

Thanks for the comment. As you said, the RH is critical factor for separating haze from fog events. Actually, the definition we used is from international definition of fog event. Fog is an observed horizontal visibility below 1000 m in the presence of suspended water droplets and/or ice crystals (NOAA, 1995), which means that when the horizontal visibility is below 1000 m, the fog events occur. Since the horizontal visibility for atmospheric haze event is usually larger than 1000 m, only the fog occurs the visibility can decrease to be less than 1000 m. So that is why we use the minimum visibility to define fog and haze events.

Theoretically, when a fog event occurs, the RH has to reach over 100 %. However, it is difficult to measure RH accurately, so in most cases, we use RH value of 90 % or 95 % as criterion to separate fog and haze. In fact, in the study region, when the RH is high enough, the fog and haze are usually co-existed. The haze aerosols can be transformed to fog droplets under certain conditions according to the Köhler curve (Köhler, 1936). It should be noticed that the situations such as heavy rain event or light fog events, which cause the horizontal visibility to be below or above 1000 m are not considered here. In addition, since we focus on long–lasting severe fog and haze event, we also include factors such as the lasting time and $PM_{2.5}$ mass concentration as additional criteria. To be more clearly, the corresponding text have been modified in the revised manuscript. We also revised Table 1 to include more parameters such as duration and maximum RH of the pollutant events, shown as below:

**Table 1: The long–lasting haze and fog–haze mixed events from January 2014 to March 2015 in Beijing city**

| Type | Starting date / time | Ending date / time | Minimum visibility (m) | Duration (h) | Maximum PM$_{2.5}$ ($\mu g\ m^{-3}$) | Maximum RH (%) | Weather phenomenon |
|---|---|---|---|---|---|---|---|
| Haze events | 2014.01.21/ 15:00 | 2014.01.24/ 15:00 | 1364 | 73 | 264 | 68 | – |
| | 2014.04.11/ 22:00 | 2014.04.14/ 23:00 | 1113 | 74 | 304 | 89 | – |
| | 2015.02.12/ 21:00 | 2015.02.16/ 10:00 | 1667 | 86 | 263 | 77 | – |
| | 2015.03.04/ 22:00 | 2015.03.08/ 10:00 | 1886 | 83 | 266 | 72 | – |
| Fog–haze mixed events | 2014.02.19/ 21:00 | 2014.02.26/ 20:00 | 647 | 168/76[b] | 269 | 92 | 02.26/16:00–21:25 Drizzle rain |
| | 2014.03.22/ 22:00 | 2014.03.28/ 14:00 | 664 | 137/13[b] | 417 | 94 | 3.28/4:30–6:20 Drizzle rain |
| | 2014.10.06/ 22:00 | 2014.10.11/ 18:00 | 500 | 117/48[b] | 391 | 100 | 10.08/6:40–7:50 10.08/10:30–11:50 Drizzle rain |
| | 2014.10.16/ 21:00 | 2014.10.20/ 23:00 | 964 | 99/3[b] | 322 | 100 | – |
| | 2014.10.22/ 4:00 | 2014.10.26/ 4:00 | 258 | 97/24[b] | 379 | 100 | – |
| | 2014.10.28/ 23:00 | 2014.11.01/ 5:00 | 837 | 79/1[b] | 184 | 100 | 10.29/23:00– 10.30/00:10 10.31/15:10–16:30 Drizzle rain |
| | 2015.01.12/ 17:00 | 2015.01.16/ 3:00 | 526 | 83/8[b] | 297 | 93[a] | 01.14/10:00–10:20 snow |

[a] the maximum RH of all valid data except missing measurements.
[b] fog–haze mixed event duration / fog duration.

4. In the context, the authors mentioned that "The PBL height derived by MPL is usually used as a reference in detecting the aerosol vertical distribution by more advanced and powerful lidars.", however, the authors also mentioned in the following paragraph that there are also some uncertainty existed for the MPL to determine the PBL height. This seems to be conflict.

**Reply 4:**

Thanks for the comment. Generally, MPL is a reliable tool to retrieval PBL height, however, it cannot work well in some situations such as that the aerosol concentration becomes uniform in the vertical direction (Tang et al., 2016). We revised this part based on your comment, as "The PBL heights retrieved by the attenuated backscatter profile of MPL and CL31 still exist some uncertainties (Tang et al., 2016; Geiß et al., 2017). Tang et al. (2016) founded that PBL height cannot be correctly obtained through sudden changes in the attenuated backscatter profiles. Such as in situation that the strong northerly winds with dry and clear air masses prevail in observation site, the atmospheric aerosols spread rapidly and became uniform in the vertical direction, the PBL height was substantially underestimated."

5. How about the statistical relationship between $PM_{2.5}$ ma concentration and PBL height from CL lidars?

**Reply 5:**

Good comment. Since MPL is much more powerful tool in retrieving PBL height, we used the data from MPL to investigate the relationship between $PM_{2.5}$ mass concentration and PBL height. When using CL data, we found that the basic relationship could be the same as that by using data from MPL, however, the correlation coefficients decreased substantially. The figure S2 below shows that the $PM_{2.5}$ concentration has inversely linear correlation with the PBL height with the $R^2$ of 0.2 for haze events and negative exponential correlation with the $R^2$ of 0.34 for fog–haze mixed events. $R^2$ are both lower than that determined by MPL. We have added these descriptions to section 3.2 in the revised manuscript and added Fig. S2 to the supplement.

[Figure]

**Figure S2: Relationship between PBL height derived by CL31 and $PM_{2.5}$ mass concentration for (a) haze and (b) fog–haze mixed events from January 2014 to March 2015 in Beijing city**

6. In this study, the authors just showed the meteorological conditions for two typical haze events? Why chose these two cases from 11 cases? The composite analysis method is suggested for the further analysis if conveniently.

**Reply 6:**

Thanks for the comment. The main reason to choose two typical cases for detailed analyses is to consider their representativeness and data completeness. This study focuses on the long–lasting haze and fog–haze mixed events, we choose typical long–lasting pollutant events and also consider whether the data for these cases is complete or not. Since the most data we used in this study were observed and supported by just a research project, it is not an easy thing to maintain these advanced instruments and obtain a complete data case from its occurrence to the end.

We have added some descriptions for this comment in the revised manuscript, as "To clarify the physical mechanism responsible for the relationship among $PM_{2.5}$, visibility and PBL height obtained above, two typical cases of long–lasting haze and fog–haze mixed events are presented and further investigated in considering their representativeness and data completeness of all cases (Table 1). In all haze events, the haze event observed from 11 to 14 April, 2014 was highly polluted with the maximum $PM_{2.5}$ concentration of 304 μg m$^{-3}$ and minimum visibility of 1113 m. For all fog–haze mixed events, the fog duration was considered firstly. Two cases are chosen, in which the fog duration accounted for more than 40 % of the total. One was observed from 19 to 26 February 2014, and the other was occurred from 6 to 11 October 2014. The maximum $PM_{2.5}$ concentration was more higher and the maximum RH reached to 100% in the fog–haze event occurred from 6 to 11 October 2014, which was chosen as typical fog–haze event for the following study."

7. The authors presented detailed analyses on the meteorological conditions for the long-lasting haze and fog-haze mixed events. However, I am still not clear what the difference for the meteorological conditions between these two cases. So, the comparison discussion in this aspect should be added in the section of Conclusion and Discussions, not just showing the common features as the current MS did.

**Reply 7:**

Thanks to the referee for the suggestion. The main differences of meteorological condition for two cases are humidity and duration. Although both cases occurred in a stable weak pressure field covering northern China, the haze event was relatively drier with shorter duration while the fog–haze mixed event was highly humid with longer duration. Since the fog droplets was formed in the fog–haze mixed event, the radiation reduction at surface was more obvious and stronger, which caused stronger descending process of the upper inversion layer, and formed a more stable PBL.

Based on your comment, we added some descriptions relevant to the differences of meteorological conditions between two cases in the section of conclusions and discussions in the revised manuscript, as "The main differences of meteorological condition for two cases are humidity and duration. Although both cases occurred in a stable weak pressure field covering northern China, the haze event was drier and duration was shorter while the fog–haze mixed event was more humid and had a longer duration. Since the fog droplets were formed in the fog–haze mixed event, the radiation reduction at surface was more

obvious and stronger, and caused stronger descending process of the upper inversion layer. In most cases, light precipitation (drizzle rain or light snow) occurred during the fog–haze mixed event while in all haze events during the observation period, there was no precipitation. The fog–haze mixed event was more favorable to form extremely high mass concentration of $PM_{2.5}$ (>300 μg m$^{-3}$) than the haze event."

8. To increase the readability, the location of Beijing is suggested to be highlighted in Figure 11 and 17.

Reply 8:

Thanks for the suggestion. The location of Beijing has been added in the Fig.S3 and Fig.S4 in the revised manuscript. The revised figures are shown as below:

[Figure]

**Figure S3: Temperature (red contour lines, units: ℃), RH (color shading, units: %) and wind (wind bar) distribution in (a) 925 hPa and (b) 850 hPa at 08:00 BST on 13 April 2014. The red dot represents the site of CMA**

[Figure]

**Figure S4: Temperature (red contour lines, units: ℃), RH (color shading, units: %) and wind (wind bar) distribution in 850 hPa at 08:00 BST on (a) 9 October and (b) 10 October 2014. The red dot represents the site of CMA.**

**3.3 Relationship between visibility and PBL height**

Theoretically, the relationship between PBL height and atmospheric visibility is not obvious under clean and clear condition. However, under polluted conditions the variation of PBL height may directly cause the variation of aerosol concentration within the PBL, and induce the change of visibility. The statistical results in Fig. 4 show that there are strong relationships between visibility and PBL height for both haze and fog–haze mixed events. A positive linear correlation with the R$^2$ of 0.35 exists in haze events and positive exponential correlation with the R$^2$ of 0.56 exists in fog–haze mixed events between visibility and PBL height.

**3.4 Physical mechanism responsible for the relationship among PM$_{2.5}$, visibility and PBL height**

To clarify the physical mechanism responsible for the relationship among PM$_{2.5}$, visibility and PBL height obtained above, two typical cases of long–lasting haze and fog–haze mixed events are presented and further investigated in considering their representativeness and data completeness of all cases (Table 1). In all haze events, the haze event observed from 11 to 14 April was highly polluted with the maximum PM$_{2.5}$ concentration of 304 μg m$^{-3}$ and minimum visibility of 1113 m. For all fog–haze

mixed events, the fog duration was considered firstly. Two cases are chosen, in which the fog duration accounted for more than 40 % of the total. One was observed from 19 to 26 February 2014, and the other was occurred from 6 to 11 October 2014. The maximum $PM_{2.5}$ concentration was more higher and the maximum RH reached to 100% in the fog–haze event occurred from 6 to 11 October 2014, which was chosen as typical fog–haze event for the following study.

**3.4.1 Typical haze event**

A typical haze event in April 2014 lasting for 74 hours starting at 22:00 LST (Local Standard Time) on 11 April and ending at 23:00 on 14 April during which visibility was less than 10000 m. The synoptic situation during the haze event characterized as a saddle field. Beijing was located in a saddle between two pairs of high and low pressure center. This weather system would lead to calm surface wind and stably stratified atmospheric condition, which was favourable for the accumulation of air pollutants. A cold front passed through Beijing on April 14 and ended the long–lasting haze event.

Figure 5 shows the temporal variations of surface meteorological and environmental factors in the whole process of the haze event. Both air temperature and RH presented a clear diurnal cycle, but they showed a gradually increasing tendency during the haze period due to the persistent southwest warm and humid airflow. The temperature increased from 7.9 ℃ to 25 ℃, with an average of 16.6±5.1 ℃, while the RH was in the range of 28 % to 89 %, with an average of 55±17 %. The variation of RH inversely corresponded that of temperature. The temperature and RH derived from PMWR showed a consistent tendency with those observed by surface automatic weather station. The wind speed varied from 0 m s$^{-1}$ to 3.9 m s$^{-1}$, with an average of 0.8 m s$^{-1}$, suggesting that the horizontal diffusion of aerosols was very weak. The $PM_{2.5}$ mass concentration was inversely correlated with visibility. $PM_{2.5}$ reached the highest at 12:00 on 14 April with the value of 304 µg m$^{-3}$, corresponding to the hourly mean visibility of 1317 m. $PM_{2.5}$ decreased dramatically after 21:00 on 14 April. The visibility continued to rise until the end of the event. The average $PM_{2.5}/PM_{10}$ is as high as 0.82, implying that fine particles were dominant in the atmosphere. The temporal variation of vertical distributions of temperature, RH, LWC and vapor density retrieved by MWRP during the whole haze event is shown in Fig. 6. Many studies demonstrated that PMWR is a useful tool to sense the thermodynamic structure of the lower troposphere continuously by providing profiles of temperature and humidity with reasonable accuracy and height resolution (Ware et al., 2003, 2013; Xu et al., 2015). The inter–comparison with the radiosonde data demonstrated the good correlation of temperature and vapor density retrievals (Guo and Guo, 2015; Xu et al., 2015). The biases of temperature retrieved by PMWR against radiosondes increased with height, and the maximum of bias is 4 ℃ under 2000 m; the bias of water vapor profile was smaller than 1 g m$^{-3}$ (Guo and Guo, 2015). Compared to the radiosonde temperature profiles in Fig. 9a, c, due to the lower vertical resolution, the PMWR could not capture the temperature inversions at the upper level. But the relatively high RH value could be well captured at the height between 700 m to 1600 m, indicating the dominant south–westerly warm and humid airflow during the haze event. The LWC was less than 0.01g m$^{-3}$.

Figure 7 shows the time–height distribution of the backscatter density detected by the CL31 and the normalized relative backscatter (NRB) of MPL, and time evolutions of the MPL–derived PBL height and $PM_{2.5}$ mass concentration during the

[revised manuscript text omitted]

Figure 10 shows the temporal evolution of surface meteorological and environmental factors in the whole process of the fog–haze mixed event. During the fog–haze mixed events, wind speed varied from 0 m s$^{-1}$ to 2.7 m s$^{-1}$, with an average of 0.5 m s$^{-1}$. The wind direction was easterly from the midnight to afternoon and then changed to calm wind until the next day morning. The temperature was in the range of 9.1 ℃ to 21.7 ℃, with an average of 15.6±3.1 ℃, while the RH was in the range of 46 % to 100 %, with an average of 88±14 %. The visibility exponentially decreased with the PM$_{2.5}$ mass concentration increasing with the R$^2$ of 0.87. The visibility decreased to the minimum 534 m in the morning of 11 October. PM$_{2.5}$ reached the highest at 19:00 on 9 October with the value of 392 μg m$^{-3}$, corresponding to the hourly mean visibility of 898 m. After that there was a slight invasion of cold air, PM$_{2.5}$ concentration decreased but still in high level. The decreasing of the temperature was favourable to the formation of fog. 
[revised manuscript text omitted]

We proposed that the PBL feedback is much stronger in fog–haze mixed event than in haze event. This is because that the fog–haze events included many fog droplets, which can substantially block the solar radiation comparing with aerosol loading in haze events in the daytime and cause stronger surface cooling. The stronger surface cooling would cause stronger descending of the upper inversion layer and then form a highly suppressed and more stable PBL. So the PBL feedback was much stronger in fog-haze event. As seen in Table 5, the radiation reduction imposed by aerosol particles was particularly stronger during the fog–haze mixed event than the haze event. The PM$_{2.5}$ concentration is higher and the PBL heights are lower in the fog–haze mixed event.

**4 Conclusion and discussions**

In this study, the relationship among PBL height, PM$_{2.5}$ mass concentration and visibility for long–lasting haze and fog–haze mixed events in Beijing was investigated and quantified. Comprehensive measurements of aerosol characteristics and meteorological conditions have been conducted in Chinese Academy of Meteorological Sciences (CAMS), Beijing since 2013, and a total of 11 long–lasting haze and fog–haze mixed events were observed from January 2014 to March 2015. PBL heights of haze and fog–haze mixed events were retrieved using MPL NRB signal and well correlated to the PBL height derived by CL31.

The statistical results show that there was a negative exponential function between the visibility and the PM$_{2.5}$ mass concentration for both haze and fog–haze mixed events with the same R$^2$ of 0.80. Aerosols could cause greater visibility impairments in fog–haze mixed events due to the increase of RH and formation of more fog drops.

The PM$_{2.5}$ concentration had inversely linear correlation with PBL height for haze events with the R$^2$ of 0.34 and negative exponential correlation with the R$^2$ of 0.48 for fog–haze mixed events, indicating that the PM$_{2.5}$ concentration is more sensitive to PBL height in fog–haze mixed events. The feedback between PBL height and PM$_{2.5}$ mass concentration became stronger when PM$_{2.5}$ mass concentration was more than 150–200 μg m$^{-3}$, particularly in fog–haze mixed cases, which is similar to the finding in haze events in Nanjing city (Petäjä et al., 2016). However, our investigation shows that this phenomenon became more obvious in  the fog–haze mixed event.

Similarity to the relationship between $PM_{2.5}$ concentration and PBL height, a positive linear correlation with the $R^2$ of 0.35 existed in haze events and positive exponential correlation with the $R^2$ of 0.56 existed in fog–haze mixed events between visibility and PBL height.

Two typical cases representing for haze and fog–haze mixed events are presented and discussed. The main results show the obvious double inversion layers located at upper–level and low–level respectively formed in both cases. The formation of upper–level inversion layer was closely associated with the persistent advection of southwest warm and humid air. The nighttime low–level inversion layer was formed due to the surface longwave radiation cooling, while the formation of daytime low–level inversion layer was complex. Our study shows that the initial daytime low–level inversion layer should be related to the surface cooling via blocking the incoming solar radiation by high aerosol loadings. The subsequent rapid development of deep low–level inversion layer in the daytime is hard to explain only by the surface cooling process. We suggest that both the heating process due to the solar radiation absorption by accumulated aerosols such as black carbon and the descended upper–level warm and humid air induced by the surface cooling played critical role in the daytime. Since the strong surface cooling could collapse PBL and cause an obviously lowing PBL height, and this process could also cause the warm and humid air at upper inversion layer to move downward and induced an increase of temperature and humidity at low air levels. The process can be clearly shown in the case of this study and it might be an important factor in strengthening low–level stability in the daytime. The positive feedback was particularly strong when the $PM_{2.5}$ mass concentration was larger than 150–200 μg $m^{-3}$. Therefore, the descended upper–level inversion layer should be an important factor in strengthening the stability in the whole PBL.

The main differences of meteorological condition for two cases are humidity and duration. Although both cases occurred in a stable weak pressure field covering northern China, the haze event was drier and duration was shorter while the fog–haze mixed event was more humid and had a longer duration. Since the fog droplets were formed in the fog–haze mixed event, the radiation reduction at surface was more obvious and stronger, and caused stronger descending process of the upper inversion layer. In most cases, light precipitation (drizzle rain or light snow) occurred during the fog–haze mixed event while in all haze events during the observation period, there was no precipitation. The fog–haze mixed event was more favorable to form extremely high mass concentration of $PM_{2.5}$ (>300 μg $m^{-3}$) than the haze event.

The new finding in this paper has important implications in explaining the frequent long–lasting polluted events in the study region. Generally, a typical pollution event is usually formed under a stable and shallow temperature–inversion condition at low atmospheric layers, and would disappear or obviously decrease when the daytime solar radiation increases. However, in the study region, we found that many severe haze and fog–haze mixed events lasted for several days even for several weeks. Most previous publications attributed the reason as the persistent abnormal weather system or high emissions. However, this study shows that except for the influence of meteorological condition and high emission, the interactions and feedbacks between PBL and aerosol loading linked by radiation process are crucial in enhancing and maintaining these polluted events. These feedbacks could cause an important variation of dynamical/thermal processes in lower troposphere. The formation of double inversion layer and their subsequent change is closely associated with persistent meteorological condition, high aerosol

loading and associated radiation process. Due to the complex interactions and feedbacks, a deeper and more stable atmospheric low–level is formed and it is hard to break up by daytime solar radiation heating process until the strong wind occurs and removes the high aerosol loading.

**Competing interests**

5    The authors declare they have no conflict of interest.

**Acknowledgements.**

[revised manuscript text omitted]

Geiß, A., Wiegner, M., Bonn, B., Schäfer, K., Forkel, R., von Schneidemesser, E., Münkel, C., Chan, K. L., and Nothard, R.: Mixing layer height as an indicator for urban air quality?, Atmos. Meas. Tech., 10, 2969-2988, doi:10.5194/amt-10-2969-2017, 2017.

Gultepe, I., Zhou, B., Milbrandt, J., Bott, A., Li, Y., Heymsfield, A. J., Ferrier, B., Ware, R., Pavolonis, M., Kuhn, T., Gurka,
10   J., Liu, P., and Cermak, J.: A review on ice fog measurements and modeling, Atmos. Res., 151, 2–19, doi:10.1016/j.atmosres.2014.04.014, 2015.

Guo, L. J., and Guo, X. L.: Verification study of the atmospheric temperature and humidity profiles retrieved from the ground–based multi–channels microwave radiometer for persistent foggy weather events in northern China, Acta Meteorologica Sinica, 73, 368–381, doi: 10.11676/qxxb2015.025, 2015. (in Chinese)

15   Guo, L., Guo, X., Fang, C., and Zhu, S.: Observation analysis on characteristics of formation, evolution and transition of a long-lasting severe fog and haze episode in North China, Sci. China Earth Sci., 58, 329–344, doi: 10.1007/s11430-014-4924-2, 2015.

Haeffelin, M., Angelini, F., Morille, Y., Martucci, G., Frey, S., Gobbi, G. P., Lolli, S., O'Dowd, C. D., Sauvage, L., Xueref-Rémy, I., Wastine, B., and Feist, D. G.: Evaluation of mixing–height retrievals from automatic profiling lidars and ceilometers
20   in view of future integrated networks in Europe, Bound.-Lay. Meteorol., 143, 49–75, doi:10.1007/s10546-011-9643-z, 2012.

Han, L., Zhou, W., and Li, W.: Fine particulate ($PM_{2.5}$) dynamics during rapid urbanization in Beijing, 1973–2013, Sci. Rep., 6, doi:10.1038/srep23604, 2016.

Han, X., Zhang, M. G., Tao, J. H., Wang, L. L., Gao, J., Wang, S. L., and Chai, F. H.: Modeling aerosol impacts on atmospheric visibility in Beijing with RAMS-CMAQ, Atmos. Environ., 72, 177–191, doi:10.1016/j.atmosenv.2013.02.030, 2013.

25   He, Q. S., Li, C. C., Mao, J. T., Lau, A. K. H., and Chu, D. A.: Analysis of aerosol vertical distribution and variability in Hong Kong, J. Geophys. Res., 113, doi:10.1029/2008JD009778, 2008.

Huang, R. J., Zhang, Y. L., Bozzetti, C., Ho, K. F., Cao, J. J., Han, Y. M., Daellenbach, K. R., Slowik, J. G., Platt, S. M., and Francesco, C.: High secondary aerosol contribution to particulate pollution during haze events in China, Nature, 514, 218–222, doi:10.1038/nature13774, 2014.

30   Kang, H. Q., Zhu, B., Su, J. F., Wang, H. L., Zhang, Q. C., and Wang, F.: Analysis of a long–lasting haze episode in Nanjing, China, Atmos. Res., 120–121, 78–87, doi:10.1016/j.atmosres.2012.08.004, 2013.

Klein, C., and Dabas, A.: Relationship between optical extinction and liquid water content in fogs, Atmos. Meas. Tech., 7, 1277–1287, doi:10.5194/amt-7-1277-2014, 2014.

Köhler, H.: The nucleus in and the growth of hygroscopic droplets, Transactions of the Faraday Society, 32, 1152-1161, doi:
35   10.1039/TF9363201152, 1936.

已移动(插入) [1]

已上移 [1]: Han, L., Zhou, W., and Li, W.: Fine particulate ($PM_{2.5}$) dynamics during rapid urbanization in Beijing, 1973–2013, Sci. Rep., 6, doi:10.1038/srep23604, 2016.

Lee, P., and Ngan, F.: Coupling of Important Physical Processes in the Planetary Boundary Layer between Meteorological and Chemistry Models for Regional to Continental Scale Air Quality Forecasting: An Overview, Atmosphere, 2, 464–483, doi:10.3390/atmos2030464, 2011.

5  Li, Q., Zhang, R. H., and Wang, Y.: Interannual variation of the wintertime fog-haze days across central and eastern China and its relation with East Asian winter monsoon, Int. J. Climatol., 36, 346-354, doi:10.1002/joc.4350, 2016.

Li, Y., Zhao, H. J., and Wu, Y. F.: Characteristics of Particulate Matter during Haze and Fog (Pollution) Episodes over Northeast China, Autumn 2013, Aerosol Air Qual. Res., 15, 853–864, doi:10.4209/aaqr.2014.08.0158, 2015.

Liao, H., Chang, W. Y., and Yang, Y.: Climatic Effects of Air Pollutants over China: A Review, Adv. Atmos. Sci. , 32, 115-139, doi:10.1007/s00376-014-0013-x, 2015.

[revised manuscript text omitted]

已移动(插入) [3]

已上移 [3]: Yang, D. W., Li, C. C., Lau, A. K. H., and Li, Y.: Long–term measurement of daytime atmospheric mixing layer height over Hong Kong, J. Geophys. Res., 118, 2422–2433, doi:10.1002/jgrd.50251, 2013.

Yang, Y., Wang, H., Smith, S. J., Ma, P. L., and Rasch, P. J.: Source attribution of black carbon and its direct radiative forcing in China, Atmos. Chem. Phys., 17, 4319-4336, doi:10.5194/acp-17-4319-2017, 2017b.

[revised manuscript text omitted]

Zou, Y., Wang, Y., Zhang, Y., and Koo, J.-H.: Arctic sea ice, Eurasia snow, and extreme winter haze in China, Sci. Adv., 3, doi:10.1126/sciadv.1602751, 2017.

**Table 1: The long–lasting haze and fog–haze mixed events from January 2014 to March 2015 in Beijing city**

| Type | Starting date / time | Ending date / time | Minimum visibility (m) | Duration (h) | Maximum $PM_{2.5}$ ($\mu g\ m^{-3}$) | Maximum RH (%) | Weather phenomenon |
|---|---|---|---|---|---|---|---|
| Haze events | 2014.01.21/ 15:00 | 2014.01.24/ 15:00 | 1364 | 73 | 264 | 68 | – |
| | 2014.04.11/ 22:00 | 2014.04.14/ 23:00 | 1113 | 74 | 304 | 89 | – |
| | 2015.02.12/ 21:00 | 2015.02.16/ 10:00 | 1667 | 86 | 263 | 77 | – |
| | 2015.03.04/ 22:00 | 2015.03.08/ 10:00 | 1886 | 83 | 266 | 72 | – |
| Fog–haze mixed events | 2014.02.19/ 21:00 | 2014.02.26/ 20:00 | 647 | 168/76[b] | 269 | 92 | 02.26/16:00–21:25 Drizzle rain |
| | 2014.03.22/ 22:00 | 2014.03.28/ 14:00 | 664 | 137/13[b] | 417 | 94 | 3.28/4:30–6:20 Drizzle rain |
| | 2014.10.06/ 22:00 | 2014.10.11/ 18:00 | 500 | 117/48[b] | 391 | 100 | 10.08/6:40–7:50 10.08/10:30–11:50 Drizzle rain |
| | 2014.10.16/ 21:00 | 2014.10.20/ 23:00 | 964 | 99/3[b] | 322 | 100 | – |
| | 2014.10.22/ 4:00 | 2014.10.26/ 4:00 | 258 | 97/24[b] | 379 | 100 | – |
| | 2014.10.28/ 23:00 | 2014.11.01/ 5:00 | 837 | 79/1[b] | 184 | 100 | 10.29/23:00– 10.30/00:10 10.31/15:10–16:30 Drizzle rain |
| | 2015.01.12/ 17:00 | 2015.01.16/ 3:00 | 526 | 83/8[b] | 297 | 93[a] | 01.14/10:00–10:20 snow |

[a] the maximum RH of all valid data except missing measurement.
[b] fog–haze mixed event duration / fog duration.

**Table 2: The exponential curve of visibility and PM$_{2.5}$ mass concentration for haze and fog–haze mixed events.**

| Function | R$^2$ | Pollutant events type |
|---|---|---|
| y=87.57495+12484.02388exp(−0.00787x) | 0.79563 | haze events |
| y=1043.96602+26206.87173exp(−0.01981x) | 0.80668 | fog–haze mixed events |

x respects the mass concentration of PM$_{2.5}$ (µg m$^{-3}$); y respects visibility (m).

[revised manuscript text omitted]

---

## Author Response (AR2)

**Manuscript # acp-2017-455**

**Reply to Co-editor**

Co-editor comments:

This study presents the analysis quantifying the relationship among $PM_{2.5}$ concentration, visibility and planetary boundary layer (PBL) height for tow haze events i.e. one long–lasting haze and one fog–haze mixed event in Beijing. Authors found negative relationships between visibility and $PM_{2.5}$ and between $PM_{2.5}$ and PBL height and suggested possible feedbacks among PM2.5, PBL, and/or humidity whereas the PBL plays a dominant role. Authors also found a double inversion layer formed in both typical events, which played critical roles in maintaining and enhancing the long–lasting polluted events. The topic of this paper is interesting and is suitable for ACP. However, major revisions are required before I can accept this manuscript for publication. Authors did a lot of analysis on the relationships among $PM_{2.5}$, visibility and PBL height. However, the findings are not new and these relationships were already reported in many previous studies. What's really new probably is the influence of double inversion layer on the meteorology and $PM_{2.5}$, which deserves more in-depth analysis and discussion. In addition, the current analysis mainly focuses on the correlations and co-changes of $PM_{2.5}$, PBL Height and visibility, more substantial analysis and discussion of the physical explanation are required for those relationships, for examples, courses and effects of those quantities as well as the role of PBL height in it.

Authors reply:

Thank you very much for your constructive comments, which are very helpful in further improving our manuscript. We have further revised our manuscript based on your two critical points regarding to the influence of double inversion layer on the meteorology and $PM_{2.5}$, and the physical explanation for those relationships we obtained in this paper. We have added more analyses and discussions in the revised manuscript. Please see the noted content in the revised manuscript.

After carefully revised, we think that the influence of double inversion layer on the meteorology and $PM_{2.5}$ as well as the physical mechanism responsible for those relationships become much clearer.

[revised manuscript text omitted]

In addition, the interactions between aerosol pollution and climate change have been substantially addressed in recent publications, for example, anthropogenic climate change (Cai et al., 2017), reduced Arctic sea ice (Wang et al., 2015; Zou et al., 2017), the Tibetan Plateau warming (Xu et al., 2016), influences of ENSO events on haze frequency in eastern China (Gao

and Li, 2015), weakened East Asian winter monsoon (Li et al., 2016), decadal weakening of winds (Yang et al., 2016), and enhanced thermal stability of the lower atmosphere (Zhang et al., 2014; Chen and Wang, 2015). The dust–wind interaction (Yang et al., 2017a) and upwind transport (Yang et al., 2017b) could also intensify haze events in China.

Fog and haze events usually occur in the stable PBL, which is located at the lowest atmospheric layer and strongly influenced by the exchange of momentum, heat, and water vapor at the earth's surface. Many previous publications showed that fog and haze events were usually formed in a weak high–pressure system with low surface wind, which was unfavourable for air mixing and pollutants diffusion (Liu et al., 2007; Kang et al., 2013; X. J. Zhao et al., 2013; G. J. Zheng et al., 2015). The aerosols directly emitted from polluted source and those secondly formed might be 
[revised manuscript text omitted]
 the previous studies, the fog–haze pollution was defined as a phenomenon in which a visibility is less than 10000 m resulted from the dense accumulation of fine aerosol particles (Fu and Chen, 2017; Liu et al., 2017; Li et al., 2016). Here, we further divided the studied cases into haze events and fog–haze mixed events. According to the Kohler curve (Köhler, 1936), the haze
5  aerosols can be transformed to fog droplets under certain meteorological conditions. The classification is usually done based on values of the visibility range (Cho et al., 2000; Tardif and Rasmussen, 2007; Elias et al., 2009). The international definition of a fog event is an observed horizontal visibility below 1000 m in the presence of suspended water droplets and/or ice crystals (NOAA, 1995). However, the fog droplets and haze aerosols are usually mixed and co-existed in the polluted region. In this study, if the fog occurred in the pollutant episode, it was defined as the fog–haze mixed event, which is similar as the study in
10  Sun et al. (2006). If the fog did not occur in the whole pollutant episode, it was defined as the haze event. Since our study focuses on long–lasting haze and fog–haze mixed events, the parameters of the $PM_{2.5}$ concentration ($>=50$ μg m$^{-3}$) and lasting–time ($>= 72$ hours) of these events were included as additional criteria. It should be noticed that the situations such as heavy rain event or light fog events, which cause the horizontal visibility to be below or above 1000 m are not considered here. From January 2014 to March 2015, the total number of persistent pollutant cases is 11, in which four haze cases and seven fog–haze
15  mixed cases were obtained. Table 1 listed all cases investigated. The RH in each haze event was lower than 90 %. The maximum RH of each haze event was 68 %, 89 %, 77 % and 72 %, respectively. The maximum RH of each fog–haze mixed event was larger than 90 %. The results were consistent with the study of Xu et al. (2016) and Q. Zhang et al. (2015), in which the observed RH less than 90 % was used to separate haze events from fog events under the visibility<10000 m due to the difficulty to measure RH correctly.

20  ## 3 Results and discussion

**3.1 Relationship between visibility and $PM_{2.5}$ mass concentration**

[revised manuscript text omitted]

The PBL heights retrieved by measuring the attenuated backscatter profile of MPL and CL31 still exist some uncertainties (Tang et al.,2016; Geiß et al., 2017). Tang et al. (2016) found that PBL height cannot be correctly obtained through sudden changes in the attenuated backscatter profiles. Such as in situation that the strong northerly winds with dry and clear air masses

prevail in observation site, the atmospheric aerosols spread rapidly and became uniform in the vertical direction, the PBL height was substantially underestimated.

The statistical relationship between $PM_{2.5}$ mass concentration and PBL height is investigated and shown in Fig. 3. It shows that the $PM_{2.5}$ concentration has inversely linear correlation with the PBL height with the $R^2$ of 0.34 for haze events and negative exponential correlation with the $R^2$ of 0.48 for fog–haze mixed events, indicating that the $PM_{2.5}$ concentration is more sensitive to the PBL height in fog–haze mixed events. The $PM_{2.5}$ concentrations of 50 µg m$^{-3}$, 100 µg m$^{-3}$, 200 µg m$^{-3}$, 300 µg m$^{-3}$, and 400 µg m$^{-3}$, correspond to the PBL heights of 830 m, 510 m, 330 m, 300 m, and 290 m, respectively in fog–haze mixed events. The $PM_{2.5}$ concentrations of 100 µg m$^{-3}$, 200 µg m$^{-3}$, 300 µg m$^{-3}$ correspond to the PBL heights of 460 m, 370 m and 280 m, respectively, in the haze events. When using CL31 data, we find that the basic relationship can be the same as that by using data from MPL, however, the correlation coefficients decrease substantially. The figure S2 shows that the $PM_{2.5}$ concentration has inversely linear correlation with the PBL height with the $R^2$ of 0.2 for haze events and negative exponential correlation with the $R^2$ of 0.34 for fog–haze mixed events. $R^2$ are both lower than that determined by MPL.

The feedback between PBL height and $PM_{2.5}$ mass concentration is obviously different in the haze events and fog–haze mixed events. In the haze events the $PM_{2.5}$ mass concentration increases almost linearly with the decrease of PBL height, while in the fog–haze mixed events the $PM_{2.5}$ mass concentration initially tends to show a relatively slow increase with the decrease of PBL height. As long as the PBL decreases to the height below 400–500 m, the slight decrease of PBL height could cause a rapid increase of $PM_{2.5}$ mass concentration. Petäjä et al. (2016) investigated the haze cases in Nanjing city and pointed out that the aerosol–boundary layer feedback remained moderate at fine particulate matter concentrations lower than about 200 µg m$^{-3}$, but that it became increasingly effective at higher particulate matter loadings. Our investigation shows that this phenomenon becomes more obvious in fog–haze mixed event.

**3.3 Relationship between visibility and PBL height**

Theoretically, the relationship between PBL height and atmospheric visibility is not obvious under clean and clear condition. However, under polluted conditions the variation of PBL height may directly cause the variation of aerosol concentration within the PBL, and induce the change of visibility. The statistical results in Fig. 4 show that there are strong relationships between visibility and PBL height for both haze and fog–haze mixed events. A positive linear correlation with the $R^2$ of 0.35 exists in haze events and positive exponential correlation with the $R^2$ of 0.56 exists in fog–haze mixed events between visibility and PBL height.

As shown above, there were strong relationships among $PM_{2.5}$ concentration, visibility and PBL height for both haze events and fog–mixed events in Beijing city. It is relatively easy to understand the relationship between $PM_{2.5}$ and visibility since the high $PM_{2.5}$ concentration can generally cause a low visibility. However, many factors can cause the increase of $PM_{2.5}$ concentration, such as the increased source emission or outside pollution transport, and the lowered PBL height as well as the formation of secondary aerosol particles as suggested by previous studies. This study shows that there was a strong negative relationship between $PM_{2.5}$ and PBL height, suggesting that the lowering PBL height might play a dominant role in the obvious

increase of PM$_{2.5}$, in particular, for high PM$_{2.5}$ condition. The critical points are how to cause the lowering process of PBL height and how to maintain these persistent polluted events, which have not been well clarified in the previous studies. In the following section, we will give two typical cases to further investigate these issues.

**3.4 Physical mechanism responsible for the relationship among PM$_{2.5}$, visibility and PBL height**

To clarify the physical mechanism responsible for the relationship among PM$_{2.5}$, visibility and PBL height obtained above, two typical cases of long–lasting haze and fog–haze mixed events are presented and further investigated in considering their representativeness and data completeness of all cases (Table 1). In all haze events, the haze event observed from 11 to 14 April was highly polluted with the maximum PM$_{2.5}$ concentration of 304 µg m$^{-3}$ and minimum visibility of 1113 m. For all fog–haze mixed events, the fog duration was considered firstly. Two cases are chosen, in which the fog duration accounted for more than 40 % of the total. One was observed from 19 to 26 February 2014, and the other was occurred from 6 to 11 October 2014. The maximum PM$_{2.5}$ concentration was much higher and the maximum RH reached to 100% in the fog–haze event occurred from 6 to 11 October 2014, which was chosen as typical fog–haze event for the following study.

**3.4.1 Typical haze event**

[revised manuscript text omitted]

5   Meteorological station during the haze event. It shows that the daily total horizontal plane direct radiation was significantly reduced by about 35 %, from 7.67 MJ m$^{-2}$ d$^{-1}$ on 13 April to 4.91 MJ m$^{-2}$ d$^{-1}$ on 14 April, and at the same time the amount of total scattering radiation was increased from 11.07 MJ m$^{-2}$ d$^{-1}$ to 12.65 MJ m$^{-2}$ d$^{-1}$. These results suggest that the reduction of surface solar radiation due to the increased aerosol loading could be an important factor in maintaining the haze event in the daytime. Generally, the strong daytime solar radiation process may break up the low–level inversion layer formed by long–

10   wave radiation cooling in the night time and dissipate the haze pollution event. However, under high pollution condition as the case studies here, the surface solar radiation could be strongly blocked and reduced by high aerosol loading in the daytime, under this condition, the inversion layer could be lifted to a higher level by the reduced solar radiation heating, but it could not be completely broken up. Thus, the haze weather could be continuous in the daytime. Moreover, the lasting inversion layer in the daytime could accumulate more and more polluted aerosols, which could cause much less surface solar radiation heating

15   and further suppress the development of PBL height, in return, produce much higher aerosol loading. This strong feedback between aerosol loading and PBL height linked by radiation process should be a main mechanism responsible for their relationship obtained above.

To further reveal the lowering process of PBL height and its impact on atmospheric stratification within the PBL in the haze event, the temperature and RH profiles and their variations are displayed in Fig. 9. It shows that the apparent features of vertical

20   temperature and RH profiles were the formation of double inversion layers and their subsequent variations during the whole haze event. The formation of the upper–level inversion layer at around 1200–1600 m should be closely associated with the warm and humid airflow from southwest. Temperature, RH and wind distribution in 925 and 850 hPa at 08:00 BST on 13 April 2014 are shown in Fig. S3, from which we could see the weak warm advection from the southwest and west at 925 hPa and 850 hPa, respectively. Relatively high RH values could be also observed at the height between 700 m and 1600 m in Fig.

25   6b. Fig. 9a also shows that the upper–level temperature inversion layer had a tendency to descend with time evolution and corresponded well to the downdraft zone displayed in Fig. 8, indicating that the PBL height tended to descend during the haze event, which is consistent with that retrieved by MPL shown in Fig. 7c. At the same time, the low–level inversion layer initially formed by surface longwave radiation cooling process in the night time on 12 April was lifted and enhanced in the daytime on 13 April and 14 April. We propose that the enhanced low–level inversion layer at around 150–600 m was primarily caused by

30   the descending process of the upper–level inversion layer at around 1200–1600 m. This can be explained by the increased temperature and humidity in the whole PBL during the haze event (Fig. 9 and Fig. 5a, b). Fig. 5a shows that the surface air temperature tended to be cooling in the night time, and warming in the daytime, but the tendency of temperature was increasing in the whole haze event, suggesting that the downdraft zone and the descending process of upper–level inversion layer might be primarily caused by the long–wave radiation cooling process in the night time, and maintained in the daytime due to the

substantially reduced surface solar radiation process. This can be verified by the vertical distribution of vertical velocities shown in Fig. 8. It shows that the downdraft zone was initially formed in the night time on 13 April and continued until at the end of haze event. In addition, the downdraft zone only occurred within a relatively narrow belt, and there were weak updrafts below it and almost no vertical winds above it, so it is hard to attribute its formation to the dominant weather system. It should be noted that the outgoing long–wave radiation cooling produced by the aerosol accumulation in the top of PBL might also contribute to the descending of the downdraft zone in the night time. But this contribution cannot be quantitively estimated due to the lack of measurement.

It is obvious that with the increased accumulation of polluted aerosols in the daytime the low–level stable layer height was increased and its stability was further strengthened. For example, the inversion layer from 150 m to 550 m with the lapse rate of air temperature is –0.38 ℃ (100 m)$^{-1}$ at 8:00 on 13 April, while the lapse rate of the same layer is –0.75 ℃ (100 m)$^{-1}$ at the same time on 14 April. Some researchers suggested that the heating process due to the solar radiation absorption by aerosols such as black carbon might play an important role in forming the deep inversion layer (Ding et al., 2016; Petäjä et al., 2016). This might be true for forming the relatively weak temperature inversion, but it is hard to explain the obvious increase of temperature and humidity in the whole vertical boundary layer in Fig. 9 and the prominent downward movement of airflow shown in Fig. 8. The descending process of the upper–level inversion layer should be responsible for the increase of temperature and humidity in the whole boundary layer and the enhancement of the low–level inversion layer, as well as the lowering PBL height in the daytime. Although the low–level inversion layer was confined to the surface due to the lack of solar radiation heating in the night time comparing with that in the daytime, the enhancement of the low–level inversion layer was still evident during the descending process of upper–level inversion layer (Fig. 9c).

In all, the haze case studied above shows that the persistent advection of southwest warm and humid air provided a long–lasting favourable condition for the formation of a stable upper–level inversion layer of PBL that weakened the upward mixing and diffusing of surface polluted aerosols in the PBL. As long as the accumulation of aerosols reached more than 150–200 μg m$^{-3}$, the low–level inversion layer initially formed by surface long–wave radiation cooling in the night time could not be broken up in the daytime due to the substantially weakening solar radiation heating induced by the high aerosol loading, so that the haze event could maintain and last until the end of haze event. At the same time, the descending process of the upper–level inversion layer induced by surface cooling process in the night time could also maintain in the daytime and enhanced the low–level inversion layer by transporting the warm and humid air to the lower levels of the PBL. The descended warm and humid air from the upper inversion layer could significantly strengthen the low–level stability and in return rapidly increased the aerosol loadings. Therefore, the formation of double inversion layers and their subsequent changes linked by radiation processes should have critical role in lowering PBL height and rapidly increasing of PM$_{2.5}$ concentration, as well as maintaining the long–lasting and severe haze weather event.

[revised manuscript text omitted]

The radiation processes were also analysed and compared for this typical fog–haze mixed event. It shows that the decrease of

15 surface radiation parameters was more obvious in the fog–haze mixed event shown in Table 4. Daily total horizontal (vertical) plane direct radiation was significantly reduced from 14.11 (24.94) to 0.86 (1.40) MJ m$^{-2}$ d$^{-1}$ during 6–11 October 2014, the reduction of horizontal (vertical) direct radiation was about 94 % in fog–haze mixed event while the total scattering radiation was increased from 3.40 to 7.21 MJ m$^{-2}$ d$^{-1}$, indicating that the daytime surface solar radiation reduction in the fog–haze mixed event was much stronger than that in the haze event. Moreover, the decrease of radiation was well corresponding to the high

20 $PM_{2.5}$ concentration recorded due to the lowering PBL height.

Similar to the haze event, the double inversion layers and variations of temperature and RH profiles could be also found and were much stronger in the fog–haze mixed event (Fig. 14). The obvious upper inversion layer was closely associated with strong advection of warm and humid airflow from southwest (Fig. S4). The relatively high RH values could be observed at the height from 500 m to 1700 m in Fig. 11b. The daily averaged PBL height reached the minimum of 270 m, and lasted until the

25 end of the fog–haze mixed event in the evening 11 October. Therefore, the enhancement of the low–level inversion layer caused by the stronger descending process of upper–level inversion layer and the lowering process of PBL height as well as the rapid increase of aerosol loadings were much obvious in the fog–haze mixed event, indicating that the interactions and feedbacks between PBL height and aerosol concentration linked by radiation process were much stronger. More stable PBL and stronger low–level inversion could be formed and induced higher aerosol loading during the fog–haze mixed event.

30 Although the absorption of light–absorbing particles such as black carbon may increase the temperature above the surface and the black carbon could contribute a fraction of about 3–15 % to the total mass concentrations in urban air (Yang et al., 2011; Huang et al., 2014), it could not be a dominant factor in forming the low–level inversion layer based on this study.

The low–level inversion layer in the night time was relatively weaker in the fog–haze mixed event than that in the haze event. This is because that the surface long–wave radiation cooling in the night time could rapidly cause the formation of fog droplets

as long as the air saturation condition was reached. During the formation of fog droplets, the condensational latent heating release could heat the air and weaken the inversion structure in the night time.

We can see that the influence of double inversion layer on the meteorology and PM$_{2.5}$ was relatively stronger in the fog–haze mixed event, but the physical mechanisms responsible for these influences were similar to those in the haze event. Table 5 summarizes the averaged PM$_{2.5}$ mass concentration, PBL height, RH and radiation parameters in the haze and fog–haze mixed event. The averaged PM$_{2.5}$ concentration was 164.5 µg m$^{-3}$ in the haze event while was 250.4 µg m$^{-3}$ in the fog–haze mixed event, corresponding to the averaged PBL heights of 470 m and 360 m, respectively. The interactions and feedbacks between PBL height and PM$_{2.5}$ concentration were much stronger due to the formation of many fog droplets in fog–haze mixed event than that in haze event. The fog droplets could substantially block the solar radiation and caused much less solar radiation heating in the daytime, so that the averaged reduction of surface solar total radiation caused by the fog–haze mixed event was almost double comparing to that by the haze event. Since the solar radiation absorbed by the earth surface may heat the bottom of atmospheric column producing convective eddies that transport heat and water vapor upward and driving the growth of the PBL (Lee and Ngan, 2011), the substantial reduction of surface solar radiation in both haze and fog–haze mixed events could induce much less surface heating and formed a much lower PBL height.

**4 Conclusion and discussions**

In this study, the relationship among PBL height, PM$_{2.5}$ mass concentration and visibility for long–lasting haze and fog–haze mixed events in Beijing was investigated and quantified. Comprehensive measurements of aerosol characteristics and meteorological conditions have been conducted in Chinese Academy of Meteorological Sciences (CAMS), Beijing since 2013, and a total of 11 long–lasting haze and fog–haze mixed events were observed from January 2014 to March 2015. PBL heights of haze and fog–haze mixed events were retrieved using MPL NRB signal and well correlated to the PBL height derived by CL31.

The statistical results show that there was a negative exponential function between the visibility and the PM$_{2.5}$ mass concentration for both haze and fog–haze mixed events with the same $R^2$ of 0.80. Aerosols could cause greater visibility impairments in fog–haze mixed events due to the increase of RH and formation of more fog drops. The PM$_{2.5}$ concentration had inversely linear correlation with PBL height for haze events with the $R^2$ of 0.34 and negative exponential correlation with the $R^2$ of 0.48 for fog–haze mixed events, indicating that the PM$_{2.5}$ concentration is more sensitive to PBL height in fog–haze mixed events. The feedback between PBL height and PM$_{2.5}$ mass concentration became stronger when PM$_{2.5}$ mass concentration was more than 150–200 µg m$^{-3}$, particularly in fog–haze mixed cases, which is similar to the finding in haze events in Nanjing city (Petäjä et al., 2016). However, our investigation shows that this phenomenon became more obvious in the fog–haze mixed event. Similarity to the relationship between PM$_{2.5}$ concentration and PBL height, a positive linear correlation with the $R^2$ of 0.35 existed in haze events and positive exponential correlation with the $R^2$ of 0.56 existed in fog–haze mixed events between visibility and PBL height.

We proposed that the PBL

The statistical results show that there were strong relationships among PM$_{2.5}$ concentration, visibility and PBL height for both haze events and fog–haze mixed events in Beijing city. In order to clarify the physical mechanism responsible for these relationships, the courses and effects of these quantities as well as the role of PBL height were further investigated based on two typical cases representing for haze and fog–haze mixed events. We found that both cases had an obvious structure of double inversion layers located at upper–level and low–level of PBL, respectively. The variations of the double inversion layers were closely associated with the processes of long–wave radiation cooling in the night time and short–wave solar radiation reduction in the daytime. The formation of upper–level inversion layer was closely associated with the persistent advection of southwest warm and humid airflow and that of low–level inversion layer was initially produced by the surface long–wave radiation cooling in the night time and continued and maintained by the substantial reduction process of surface solar radiation in the daytime. The obvious descending process of the upper–level inversion layer could be responsible for the enhancement of the low–level inversion layer and the lowering PBL height, as well as high aerosol loading for these polluted events. The descending process of the upper–level inversion layer was initiated by long–wave radiation cooling in the night time and continued and maintained by the substantial reduction process of surface solar radiation in the daytime. Therefore, the variations of all these quantities were closely linked and driven by different radiation processes and the change of the double inversion layers was closely related to the lowering PBL height and high PM$_{2.5}$ concentration. The reduction of surface solar radiation in the daytime could be around 35 % for haze event and 94 % for fog–haze mixed event.

The descending process of upper–level inversion layer could cause the warm and humid airflow from southwest to move downward and induced an increase of temperature and humidity in the whole PBL, and formed a deeper and more stable PBL. All these processes can be clearly shown in the typical cases of this study. The descending of the upper–level inversion and its subsequent interactions and feedback with quantities such as low–level inversion layer, PBL height and PM$_{2.5}$ concentration were particularly strong when PM$_{2.5}$ mass concentration was larger than 150–200 µg m$^{-3}$.

The main differences of meteorological condition for two cases are humidity and duration. Although both cases occurred in a stable weak pressure field covering northern China, the haze event was drier and duration was shorter while the fog–haze mixed event was more humid and had a longer duration. Since the fog droplets were formed in the fog–haze mixed event, the radiation reduction at surface was more obvious and stronger, and caused stronger descending process of the upper inversion layer. In most cases, light precipitation (drizzle rain or light snow) occurred during the fog–haze mixed event while in all haze events during the observation period, there was no precipitation. The fog–haze mixed event was more favorable to form extremely high mass concentration of PM$_{2.5}$ (>300 µg m$^{-3}$) than the haze event. The daily averaged PBL height was lower than 110 m in the fog–haze mixed event. The relationships among PM$_{2.5}$, visibility and PBL height in both typical cases are found to be consistent with those obtained by statistical analyses.

The new finding in this paper has important implications in explaining the frequent long–lasting polluted events in the study region. Generally, a typical pollution event is usually formed under a stable and shallow temperature–inversion condition at low atmospheric layers, and would disappear or obviously decrease when the daytime solar radiation increases. However, in the study region, we found that many severe haze and fog–haze mixed events lasted for several days even for several weeks.

Most previous publications attributed the reason as the persistent abnormal weather system or high emissions. However, this study shows that except for the influence of meteorological condition and high emission, the interactions and feedbacks between meteorological factors and aerosols were crucial. We found that the formation of double inversion layers in the PBL and their subsequent variations linked by different radiation processes were crucial in enhancing and maintaining these polluted events, which could cause an important variation of dynamical/thermal processes in lower troposphere. Due to the complex interactions and feedbacks, an atmospheric environment with much deeper and more stable PBL could be formed and it is hard to break up by daytime solar radiation heating process until the strong wind occurs and removes the high aerosol loading.

**Competing interests**

The authors declare they have no conflict of interest.

**Acknowledgements.**

This research was supported by the Research and Development Special Fund for Public Welfare Industry (Meteorology) (GYHY200806001, GYHY201306047 and GYHY201406001), the National Natural Science Foundation of China (41605111), the Chinese Academy of Meteorological Sciences Basic Research and Operation Fund (2016Z004).

**Table 1: The long–lasting haze and fog–haze mixed events from January 2014 to March 2015 in Beijing city**

| Type | Starting date / time | Ending date / time | Minimum visibility (m) | Duration (h) | Maximum PM$_{2.5}$ ($\mu g \, m^{-3}$) | Maximum RH (%) | Weather phenomenon |
|---|---|---|---|---|---|---|---|
| Haze events | 2014.01.21/ 15:00 | 2014.01.24/ 15:00 | 1364 | 73 | 264 | 68 | – |
| | 2014.04.11/ 22:00 | 2014.04.14/ 23:00 | 1113 | 74 | 304 | 89 | – |
| | 2015.02.12/ 21:00 | 2015.02.16/ 10:00 | 1667 | 86 | 263 | 77 | – |
| | 2015.03.04/ 22:00 | 2015.03.08/ 10:00 | 1886 | 83 | 266 | 72 | – |
| Fog–haze mixed events | 2014.02.19/ 21:00 | 2014.02.26/ 20:00 | 647 | 168/76[b] | 269 | 92 | 02.26/16:00–21:25 Drizzle rain |
| | 2014.03.22/ 22:00 | 2014.03.28/ 14:00 | 664 | 137/13[b] | 417 | 94 | 3.28/4:30–6:20 Drizzle rain |
| | 2014.10.06/ 22:00 | 2014.10.11/ 18:00 | 500 | 117/48[b] | 391 | 100 | 10.08/6:40–7:50 10.08/10:30–11:50 Drizzle rain |
| | 2014.10.16/ 21:00 | 2014.10.20/ 23:00 | 964 | 99/3[b] | 322 | 100 | – |
| | 2014.10.22/ 4:00 | 2014.10.26/ 4:00 | 258 | 97/24[b] | 379 | 100 | – |
| | 2014.10.28/ 23:00 | 2014.11.01/ 5:00 | 837 | 79/1[b] | 184 | 100 | 10.29/23:00– 10.30/00:10 10.31/15:10–16:30 Drizzle rain |
| | 2015.01.12/ 17:00 | 2015.01.16/ 3:00 | 526 | 83/8[b] | 297 | 93[a] | 01.14/10:00–10:20 snow |

[a] the maximum RH of all valid data except missing measurement.
[b] fog–haze mixed event duration / fog duration.

**Table 2: The exponential curve of visibility and PM$_{2.5}$ mass concentration for haze and fog–haze mixed events**

| Function | R$^2$ | Pollutant events type |
|---|---|---|
| y=87.57495+12484.02388exp(–0.00787x) | 0.79563 | haze events |
| y=1043.96602+26206.87173exp(–0.01981x) | 0.80668 | fog–haze mixed events |

x respects the mass concentration of PM$_{2.5}$ (µg m$^{-3}$); y respects visibility (m).

[revised manuscript text omitted]